# Quantifying the Impact of Modeling Fidelity on Different Substructure Concepts - Part II: Code-to-Code Comparison in Realistic Environmental Conditions

Francesco Papi[1], Giancarlo Troise[2], Robert Behrens de Luna[3], Joseph Saverin[3], Sebastian Perez-Becker[3], David Marten[3], Marie-Laure Ducasse[4], Alessandro Bianchini[1]

[1]Department of Industrial Engineering, University of Florence, Firenze, 50139, Italy
[2]Seapower scrl, Naples, 80121, Italy
[3]Hermann Föttinger Institute, Technical University of Berlin, Berlin, 10623, Germany
[4]Saipem S.A., 1/7 Avenue San Fernando, 78884 Saint Quentin Yvelines cedex, France

*Correspondence to*: A. Bianchini (alessandro.bianchini@unifi.it) or F. Papi (fr.papi@unifi.it)

**Abstract.** Floating offshore wind is widely considered as a promising technology to harvest renewable energy in deep ocean waters and increase clean energy generation offshore. While evolving quickly from a technological point of view, Floating Offshore Wind Turbines (FOWTs) are challenging, as their performance and loads are governed by complex dynamics that are a result of the coupled influence of wind, waves, and currents on the structures. Many open challenges therefore still exist, especially from a modeling perspective. This study contributes to the understanding of the impact of modeling differences on FOWT loads by comparing three FOWT simulation codes, QBlade-Ocean, OpenFAST, and DeepLines Wind® and three substructure designs, a semi-submersible, a spar-buoy, and the two-part concept Hexafloat in realistic environmental conditions. This extensive comparison represents one of the main outcomes of the H2020 project FLOATECH. In accordance with international standards for FOWT certification, multiple design situations are compared, including operation in normal power production and parked conditions. Results show that the compared codes agree well in the prediction of the system dynamics, regardless of the fidelity of the underlying modeling theories. Some differences between the codes emerged however in the analysis of fatigue loads, where, contrary to extreme loads, specific trends can be noted. With respect to QBlade-Ocean, OpenFAST was found to overestimate lifetime damage equivalent loads up to 14%. DeepLines Wind®, on the other hand, underestimated lifetime fatigue loads by up to 13.5%. Regardless of the model and FOWT design however, differences in fatigue loads are larger for tower base loads than for blade root loads, due to the larger influence substructure dynamics have on these loads.

## 1 Introduction

In recent years industrial and academic interest around floating offshore wind energy has been increasing, thanks to its promise to foster wind energy harvesting in offshore areas previously inaccessible with bottom-fixed wind turbines. To fully exploit the advantages of this technology, ever larger and more flexible offshore turbines are being developed and deployed. These systems are challenging to model, as their dynamics are governed by the coupled influence of aerodynamics, hydrodynamics, control,

and moorings. As an additional complexity, with large and flexible turbine rotors, aeroelastic coupling also plays an important role. Many of the industry's work-horse simulation codes have been developed with smaller, more rigid, bottom-fixed rotors in mind and rely on engineering models, sometimes empirically derived, to model the relevant physical phenomena. In this context, a real need for verification and validation of these tools exists. Several efforts, past and present, have been put into verification and validation of offshore simulation codes. Notable examples being the Offshore Code Comparison ("OC" in short) programs promoted by the International Energy Agency (IEA), OC3, OC4, OC5, OC6 (Jonkman and Musial, 2010; Robertson et al., 2014b, 2017; Bergua and et. al., 2023) and the on-going OC7. Throughout the OC- projects, offshore codes have been compared against other codes, and against wave-tank experiments. Especially OC4 and OC5 have helped highlight deficiencies in low-frequency hydrodynamic modeling of semi-submersible type platforms (Robertson et al., 2017) that have allowed the advance of the state-of-the art in OC6 (Robertson et al., 2020; Wang et al., 2022). Most of these campaigns have found that even simplified engineering tools are generally able to capture the aerodynamics of these systems well - at times better than expected, such as in (Bergua and et. al., 2023) – when compared to higher-fidelity and more physically complete aerodynamic models. Throughout these comparison studies however, a limited number of often simplified inflow conditions have been tested. On the other hand, some authors have found some differences between modeling theories when the coupled system dynamics are put to the test. In particular, Corniglion, (2022) found increased blade root fatigue loads when comparing Blade Element Momentum Theory (BEMT) to a higher fidelity Lifting-Line Free Vortex Wake (LLFVW) method. Similar considerations were also drawn by other authors such as (Boorsma et al., 2020; Perez-Becker et al., 2020) when comparing fatigue load predictions on onshore wind turbines. In detail, Boorsma et al. (Boorsma et al., 2020) have linked the increase in fatigue loads to increased 1P load variation, while Perez-Becker et al. (Perez-Becker et al., 2020) have found that even small differences in aerodynamic modeling can lead to different controller reactions, influencing overall loading and highlighting the importance of accurately modeling the entire coupled dynamics of the system. In the case of FOWTs, dynamics are even more complex as the turbine moves in response and in reaction to the incoming wind and wave variations. This introduces additional inertial and gravitational loading on many structural components (Jonkman and Matha, 2011). Thus, differences in rotor loading may influence the response of the system, indirectly influencing other component loads and amplifying the differences between the models.

The current study contributes to the field by presenting the outcomes of an extensive code-to-code comparison considering realistic environmental conditions and three different floating substructure designs. Environmental conditions from an existing European site are obtained using the procedure described in (Papi et al., 2022c) to obtain realistic distributions of wind speed, significant wave height, peak spectral period and wind-wave misalignment. The three test-cases - a spar-buoy, a semi-submersible and the innovative two-part floater concept; Hexafloat, recently proposed by Saipem - are simulated in a variety of Design Load Cases (DLCs), including both power-production and parked conditions, as well as wind gusts. The test-cases are simulated using three offshore codes, OpenFAST, DeepLines Wind and QBlade-Ocean, which was recently extended to enable offshore simulations within the Horizon 2020 project FLOATECH. The latter code includes higher-fidelity modeling features such as LLFVW wake aerodynamics and explicit buoyancy calculation, as illustrated in (Behrens De Luna et al., 2023).

The predicted dynamics are compared in terms of extreme loads, fatigue loads and statistics. Time series are also compared in detail to give more insight into the differences in dynamics. The entire input conditions and compared datasets are available open-access and can act as validation databases for other offshore codes or as a benchmark for future modeling improvements. An extensive comparison, involving three different models with different substructure designs, three different numerical codes and multiple DLCs that include hundreds of simulations is an important point of novelty of this study and does not come without challenges. In fact, comparing coupled simulations that are aero-hydro-servo-elastic in nature such as in this study makes isolating the potential sources of any differences challenging. Nonetheless, it offers the unique opportunity of evaluating the trade-off between computational time and accuracy of the modeling theories in terms of their impact on the final design load predictions in a realistic scenario. It also allows one to highlight user-bias in the set-up of FOWT simulations. In this view, some critical aspects to consider during model set-up, that lead to significant differences in ultimate and fatigue loads in the compared models such as structural damping ratios and control strategy are discussed in detail. Ultimately, the objective of this work is to provide wind turbine modelers and practitioners with a quantitative indication of the impact that model fidelity has on FOWT design loads and provide guidance in the selection of the most suitable approach for each task at hand.

This paper is organized as follows: In Sect. 2 the procedure required to set up the code-to-code comparison that is presented herein is detailed, starting from environmental conditions and continuing with DLC definition, test-case selection, and data post-processing. In Sect. 3 some details regarding the modeling theories underpinning the compared tools are given. In Sect. 4 the main results are presented, starting from a general statistical comparison of key metrics, and then moving to the comparison of design-driving extreme and fatigue loads. The principal results are discussed, and the conclusions drawn in Sect.5

## 2 A Procedure for Code-to-Code Comparison of FOWTs in Realistic Environmental Conditions

The set-up of a design load calculation of a FOWT is a complex task on its own. Expertise is required in the selection and set-up of relevant DLCs in compliance with the various international standards (International Electrotechnical Commission, 2019a; DNVGL, 2016). In the case of FOWTs, expertise is also required in the selection of environmental conditions to use, which are site dependent. Finally, a full load calculation can produce thousands of hours of time series data, and data processing techniques are required to make it more manageable and useful for the design process. In the context of this study, all these aspects will be briefly presented as they have already been touched upon in two publications by the authors (Papi et al., 2022c; Papi and Bianchini, 2023), that will be referenced later on in this Section where appropriate.

### 2.1 European Met-Ocean Conditions

Design classes are not currently prescribed for any type of offshore wind turbine as they are for onshore wind turbines, in favor of standardization. Although the need for such standardization is acknowledged and encouraged in the DNVGL-SST-0119 design standard (DNVGL, 2018), the designer is currently required to verify the turbine and substructure combination of choice for specific installation sites. As discussed in the following Sections, standards require the definition of specific wind conditions,

normally grouped in "models" such as the Normal Turbulence Model (NTM), and sea condition, generally grouped in "sea states". Some databases containing such met-ocean data can be found in previous work – for a comprehensive literature review see (Papi and Bianchini, 2023) – however if we restrict our research to Europe, we did not find met-ocean conditions that were completely suitable for this analysis. In fact, although conditions for some reference European sites can be found in the open literature, such as in (Li et al., 2015), specific environmental contours are required to perform the ultimate load calculations according to the prescriptions of International Standards. Therefore, an open-source procedure to obtain and prepare long-term environmental data so it can be used in a design load calculation of an offshore wind turbine was developed. The procedure that is detailed in (Papi et al., 2022c) and is available open-access for others to use and improve upon (10.5281/zenodo.6972014). A highlight of the procedure is the fact that the statistical description of the installation site also includes wind-wave misalignment, which has been shown to have a significant effect on loading (Stewart, 2016).

Data is obtained from the Copernicus re-analysis database ERA5. Environmental data is available on a 30x30 km grid, therefore the procedure can be applied to a generic world-wide offshore site. In this study, hourly records of wind speed, wind direction, significant wave height, wave direction and peak spectral period from 1979 to 2000 for a site located west of the Scottish island of Barra are used. This location was chosen because of its particularly harsh environment, expected to increase non-linearities and differences in the examined models, and because it is also used in other EU-funded projects such as LifeS50+ (Antonia Krieger et al., 2015) and CoreWind (Vigara et al., 2020). Although more research would be needed to properly support this claim, due to the severity of the considered met-ocean conditions, it is reasonable to believe that any differences between the codes represent an upper limit, and smaller differences are likely to be found in less demanding conditions.

The open-source Python tool Virocon (Haselsteiner et al., 2019) is leveraged to build a joint probabilistic model of the dataset, able to describe the long-term probability of the four environmental variables that are considered: wind speed ($U_w$), significant wave height ($H_S$), peak spectral period ($T_P$) and wind-wave misalignment ($M_{ww}$). The model is then used to find the most likely combination of $H_S$ and $T_P$ for a given $U_w$, defining the Normal Sea State (NSS), and to define environmental contours: extreme conditions with 50-year recurrence period that are used to define the Extreme Sea State (ESS) and the Severe Sea State (SSS). More details on how these sea states are defined are summarized in (Papi et al., 2022c), while information on environmental contours and their applications to offshore wind turbines can be found in (Haselsteiner et al., 2020, 2021; Valamanesh et al., 2015).

**2.2 DLC Selection and Simulation Conditions**

Code-to-code comparisons in a variety of environmental conditions are performed in this study. As such, simulations in various met-ocean conditions are performed. The specific combination of met-ocean condition and operating condition is a Design Load Case (DLC). In this study normal operating conditions and parked DLCs are simulated, as shown in Table 1. , While this paragraph contains a general overview of the selected DLCs, a more detailed explanation of the selection process can be found in the FLOATECH project deliverables (Papi et al., 2022a, b), and in (Papi and Bianchini, 2023). To obtain representative ultimate loads, operation in extreme turbulence (DLC 1.3), in severe seas (DLC 1.6) and during an extreme operating gust with

direction change (DLC 1.4) are considered. In these load cases, wind and waves are considered aligned as a worst case scenario, in compliance with international standard prescriptions (International Electrotechnical Commission, 2019b). In DLCs where the turbine is parked during one year (DLC 6.3) and fifty years extreme environmental conditions, with (DLC 6.2) and without (DLC 6.1) grid loss, a $\mp 30°$ wind-wave misalignment is also considered. All ultimate load DLCs simulations are one hour long, with the exception of DLC 1.4, where simulations are 10 minutes long. In this DLC, interest is put on the extreme loads caused by the transient wind gust. As such, these simulations can be shortened without loss of relevant information. Moreover, multiple turbulent seeds and yaw misalignments are considered within each DLC. For fatigue loads, normal operation in normal inflow and sea conditions (DLC 1.2) is considered. In this DLC, in accordance with indications coming from design standards (International Electrotechnical Commission, 2019a), that require the full design space to be explored, multiple sea states are examined, including multiple combinations of the four environmental variables. Therefore, the design space is divided into bins, and at least one model evaluation for each bin is required. To keep the number of simulations manageable in the context of a code-to-code comparison endeavor, two strategies to reduce the number of required model evaluations are adopted. Both strategies were proposed in (Stewart, 2016); the first is the "probability sorting method", where the least likely bins are discarded as these conditions are unlikely and are expected to have little impact on fatigue loads. In this study the most likely bins, ensuring a total combined probability of 90% are kept in the analysis. The second strategy is bin coarsening, in which the width of the bins is increased, thereby reducing their number. As discussed in (Papi and Bianchini, 2023), by combining the two strategies a relatively manageable number of bins is obtained: 252. For each bin two half-hour simulations are performed with different yaw misalignments. The half-hour simulation length differs from the more commonly used one or three-hour simulation lengths. The rationale for such long simulations is to allow for enough time for low-frequency response, typical of FOWT systems, to build-up. Existing research (Stewart, 2016) however, indicates that the total time that is simulated within each environmental bin is the most important factor for fatigue-load estimation, rather than the length of each simulation. Moreover, based on the results in (Stewart, 2016), increasing simulation time beyond half-hour for each environmental bin does not appear to yield improved fatigue estimations in most cases. Therefore, considering the comparative nature of the study, two half-hour simulation for each environmental bin were considered sufficient for fatigue load comparison.

Table 1: DLCs used in this study. Normal operating conditions in various sea states and turbulence levels in DLCs 1.2 to 1.6 for the evaluation of fatigue (F) and ultimate (U) loads. In DLCs 6.1 to 6.3 the FOWTs are parked in extreme conditions. In DLC 6.2 a grid loss scenario is modelled, and thus multiple values of yaw-error are considered. Acronyms and abbreviations are described in nomenclature list.

| DLC | wind | | waves | | | | dur. [s] | seeds/ws | yaw | n° ws | sims | type |
| --- | --- | --- | --- | --- | --- | --- | --- | --- | --- | --- | --- | --- |
| | model | speed | model | height | period | dir. | | | | | | |
| **1.2** | NTM | $V_{in}$-$V_{out}$ | NSS | - | - | MUL | 1800 | 1 | 0, 10° | 11 | 504 | F |
| **1.3** | ETM | $V_{in}$-$V_{out}$ | NSS | $E[H_S\|V_{hub}]$ | $E[T_P\|H_S]$ | COD | 1800 | 9 | 0, $\mp 10$ | 11 | 99 | U |
| **1.4** | ECD | $V_r$ $\mp 2$ m/s | NSS | $E[H_S\|V_{hub}]$ | $E[T_P\|H_S]$ | COD | 600 | - | 0 | 6 | 12 | U |

| 1.6 | NTM | $V_{in}$-$V_{out}$ | SSS | $H_S$, SSS | $E[T_P|H_S]$ | COD | 3600 | 9 | $0, \mp10$ | 11 | 99 | U |
|-----|-----|-----|-----|-----|-----|-----|-----|-----|-----|-----|-----|-----|
| 6.1 | EWM50 | $V_{50}$ | ESS | $H_S50$ | $E[T_P|H_S]$ | $0°, \mp30°$ | 3600 | 2 | $0, \mp10$ | 1 | 12 | U |
| 6.2 | EWM50 | $V_{50}$ | ESS | $H_S50$ | $E[T_P|H_S]$ | $0°, \mp30°$ | 3600 | 2 | 0,45,90 135,180 | 6 | 12 | U |
| 6.3 | EWM1 | $V_1$ | ESS | $H_S1$ | $E[T_P|H_S]$ | $0°, 30°$ | 3600 | 2 | $0, \mp20$ | 1 | 12 | U |

To ensure a fair comparison between the codes an attempt was made to match environmental inputs as well as possible in the numerical models. The wave time series are generated in DeepLines and then imported in OpenFAST and QBlade, while the wind fields are generated by each participating institution using the same TurbSim (Jonkman, 2014) settings. The same wind fields are used in all three test cases, as if they were installed in the same site, regardless of the rotor size used. Therefore, the larger 10MW rotor defines the overall size of the wind field. A schematic representation of the wind fields is shown in Fig. 1.

**Figure 1: Schematic illustration of the wind field dimensions as used in this study with respect to the NREL 5MW and DTU 10MW rotors. The same wind fields are used on all three test-cases regardless of rotor size.**

## 2.3 Considered FOWT Designs

For the sake of generality and completeness of the analysis three floating turbine concepts are analyzed. Each test case features a different floating platform concept, namely a semi-submersible, a spar-buoy and Hexafloat. The three concepts are all derived from those in (Perez-Becker et al., 2022; Behrens De Luna et al., 2023), where some calibration was required to properly align the models with the experiments. The main characteristics of the three test-cases are detailed in the following.

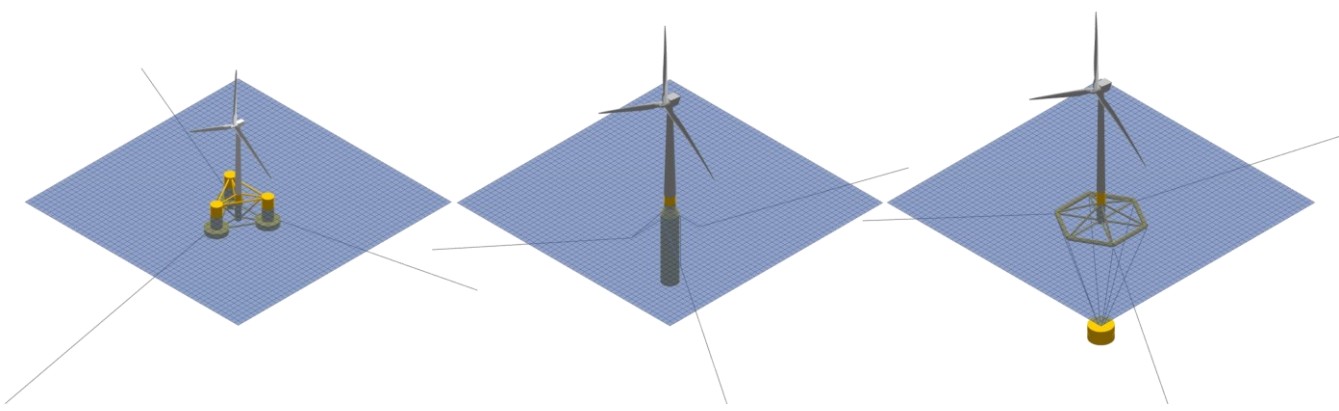

**Figure 2: Illustration of the examined numerical models in QBlade-Ocean. From left to right: NREL 5MW OC4, DTU 10MW Softwind**
**and DTU 10MW Hexafloat.**

### 2.3.1 NREL 5MW OC4 DeepCwind

The NREL 5MW OC4 semi-submersible FOWT (hereafter OC4) is an open-access turbine model defined in (Robertson et al., 2014a), upon which many code-to-code comparison exercises are based (Robertson et al., 2014b, 2017). It makes use of the NREL 5MW RWT rotor (Jonkman et al., 2009), representative of a utility-scale multi-MW rotor. The rotor is mounted on the DeepCwind semisubmersible floating platform. The platform was developed with the aim of generating test data for use in the validation of FOWT modeling tools.

The same tower design that was developed for use on the OC3-Hywind spar platform (Jonkman, 2010) is used. The semi-submersible floater consists of a main central column connected to the tower and three side columns spaced 120° apart. The offset columns are larger at the base, acting like heave plates to control the vertical motion of the FOWT and are connected together through a series of braces. A catenary mooring system is used. Three 120° lines are used to anchor the turbine to the seabed with one mooring line pointing directly upwind and the other two downwind.

### 2.3.2 DTU 10MW Softwind

The DTU 10 MW Softwind spar FOWT (hereafter Softwind) is a 1:40 scale floating platform designed by École Centrale de Nantes to develop, demonstrate, and validate a Software in the Loop (SiL) approach whereby an actuator is used to simulate the aerodynamic forcing at model scale in place of a scaled rotor. The model and experiments are described in (Arnal, 2020). The rotor nacelle assembly (RNA) is described in (Bak et al., 2013). With respect to the models used in (Behrens De Luna et al., 2023) that mimic the characteristics of the experiments (Arnal, 2020), some changes were implemented to increase the robustness of the numerical simulations when using the realistic met-ocean conditions considered in this work. Namely, the tower was

stiffened, moving to a stiff-stiff design to avoid wave and 3P tower resonance. The tower designed by Olav-Olsen[1] in the
LifeS50+ project for the OO-Star floater is used (Borg, 2015; Yu et al., 2018). Notably this tower is heavier than the one used
in the Softwind test campaign. The mass distribution in the floater is also changed. In order to have a realistic mass distribution
and inertial properties, we hypothesized the use of high-density ballast in the spar body, thus lowering the Center of Gravity
(CoG) with respect to the scaled model used in the experiments, which housed control electronics and batteries within the buoy.
The mass of the floater is also lowered by approximately 2% to compensate for the heavier tower and maintain approximately
the same draft. Furthermore, lowering the CoG lowers the platform pitch natural period, allowing for the use of a faster controller,
as explained in Sect. 3.3. The specific changes are detailed in (Papi et al., 2022a). This modified floater design is not intended
to be built and is only meant for numerical comparisons using a realistic design that is also numerically stable. These changes
are therefore deemed appropriate for the goal of this study.
In DeepLines, after unsuccessful initial attempts to align the model to QBlade and OpenFAST, and, in an initial phase, to the
Softwind experiments (Arnal, 2020), a different tuning approach was employed for the hydrodynamics of the model. In
particular, the pitch and roll inertias of the floater were decreased to better align the respective natural frequencies in free decay
tests, and additional added mass on the spar buoy was introduced through Morison's equation to improve the agreement during
surge free-decay tests. Lastly, mooring line tension was lowered to better align with the experimental data. A full description of
the differences can be found in (Papi et al., 2023).

### 2.3.3 DTU 10MW Hexafloat

The DTU 10MW Hexafloat FOWT consists of the DTU 10MW RWT mounted to the Hexafloat floater concept proposed by
Saipem. As shown in Fig. 2, the substructure consists of a floater made of relatively slender steel braces connected to a
counterweight by six tendons. This floater configuration did not require changes to the tower design and therefore the standard
onshore tower of the DTU 10MW RWT (Bak et al., 2013) is used. This model is in effect identical to the one used and described
in (Perez-Becker et al., 2022; Behrens De Luna et al., 2023)

### 2.4 Post-Processing and Data Management

The raw time series data obtained for the three models is post-processed using open-source tools, namely MLife (Hayman, 2012)
and MExtremes (Buhl, 2015) developed by NREL. The main sensors that are compared in the study are shown in Tab. 1 and
consist of blade root and tower base bending moments, mooring line fairlead tensions, nacelle fore-aft acceleration, control
signals and platform motions. Some of these sensors act like a proxy to compare the influence of various physical phenomena
on loads, such as nacelle acceleration that is used to gauge inertial loads on the tower and platform pitch that is used as indication

---

[1] The OO-Star Wind Floater has been developed by Dr. Techn. Olav Olsen (OO) since 2010 and is the property OpenFAST Floating Wind Solutions AS. OO has approved that the public model from LifeS50+ can be used for the research activities within FLOATECH. The model shall not be used for other purposes unless it is explicitly approved by OO.

of gravitational tower loading. The mechanisms that relate platform motions and substructure loading are discussed in (Robertson and Jonkman, 2011; Papi and Bianchini, 2022) and will only briefly be explained throughout this work where necessary.

MLife is used to compute Damage Equivalent Loads (DELs). DELs are the cyclic load amplitudes that cause the same fatigue damage to the structure over a certain number of cycles as the time series of a given load sensor. The Palmgren-Miner linear damage accumulation hypothesis is used to derive DELs, which can therefore only be considered representative equivalent loads if this hypothesis is valid. In this study zero-mean DELs are considered, and thus the mean of each loading cycle is disregarded. 1Hz DELs give the equivalent damage during one simulation, while lifetime DELs represent the equivalent damage over the entire lifetime of the turbine. They can be conceptually thought of as a combination of 1Hz DELs weighted by their respective probability of occurrence, which in this case is a distribution that depends on the four environmental variables defined in Sect. 2.1. As shown in Tab. 1, only the simulations in DCL 1.2 are used to compute DELs.

MExtremes is used to compute ultimate loads on the structure. In this case, DLCs 1.3, 1.4, 1.6, 6.1, 6.2 and 6.3 are used. To obtain a conservative estimate of ultimate loads in accordance with IEC 61400-1 annex I (International Electrotechnical Commission, 2019a), an averaging approach is used when computing ultimate loads, as explained in (Buhl, 2015).

**Table 1: Sensors considered in the analysis.**

| Sensor | OpenFAST ref. sys. | Name | Type |
|---|---|---|---|
| Blade root in-plane/out-of-plane bending moment | Coned CS **c** | B# Mx / B# My | F/U |
| Tower base fore-aft/side-side bending moment | Tower base CS **t** | TB My/TB Mx | F/U |
| Mooring line fairlead tensions | - | T ML# | F/U |
| Nacelle fore-aft acceleration | Tower top CS **p** | Nac. TAx | U |
| Control signals (blade pitch, gen. torque, rotor speed) | - | $\theta, \tau, \Omega$ | - |
| Platform motions (computed @SWL) | Platform CS | surge, sway, pitch, etc… | - |

## 3 Methods

This work leverages some of the authors' past experience and as such many of the same modeling techniques as described in (Behrens De Luna et al., 2023) are used, where a more complete description of the employed methods can be found. Three distinct numerical tools are used in this code-to-code comparison: OpenFAST v3.0, DeepLines Wind® and QBlade-Ocean. The tools have been compared to experimental results on scaled models and have shown, after adequate model tuning, good ability to capture the behavior of the different systems. The results of this modeling and validation effort are discussed in (Perez-Becker et al., 2022; Behrens De Luna et al., 2023). The main numerical models in each code are described in this Section.

### 3.1 Aerodynamic Models

All the models compared herein use low- to medium-fidelity aerodynamic models. The blade aerodynamics are not explicitly modeled. Instead, a series of 2D aerodynamic coefficients is used in their place. Corrections to account for 3D flow effects are

built into the aerodynamic coefficients for all the models. Moreover, Gonzalez's variant of the Beddoes-Leishman dynamic stall model (Leishman, 2016; Damiani and Hayman, 2019) is used in OpenFAST. In QBlade dynamic stall is modeled using Øye's model (Marten, 2020), while in DeepLines no unsteady airfoil aerodynamics are accounted for. The relative velocities acting on the blades are determined by the wake model. A Dynamic Blade Element Momentum (DBEM) wake model is used in OpenFAST and DeepLines, where the rotor is divided into a series of radial and azimuthal streamtubes and for each streamtube a momentum balance is performed. More details on BEM models can be found in (Burton, 2001; Hansen, 2008), and details regarding the specific DBEM model implemented in OpenFAST are in (Ning et al., 2015; Branlard et al., 2022). These models have been the industry workhorse for decades and although very simple, they have been extended in time through the addition of empirical sub-models and now fully qualify as engineering models. A higher-order Lifting Line Free Vortex Wake (LLFVW) model is used in QBlade. Here, the wake is modeled as a series of vortex filaments. Wake nodes are advected downstream by the incoming wind speed and the cumulative induction of all wake filaments. More details on these models and how they are implemented in QBlade can be found in (Van Garrel, 2003; Marten et al., 2015). The same aerodynamic lift and drag tables are used in all three codes for both aerodynamic models and correspond to the public definitions of the NREL 5MW and DTU 10MW rotors.

**3.2 Structural Models**

Structural dynamics are modeled with a modal-based linear superposition approach in OpenFAST through the submodule ElastoDyn. One limitation is that blade torsion is not modeled in ElastoDyn. In QBlade and DeepLines on the other hand, a higher fidelity finite-element approach is used, whereby the structural dynamics are modeled with a multi-body representation that uses Euler-Bernoulli beam elements in a co-rotational formulation (Marten, 2020; Le Cunff et al., 2013). Within OpenFAST a more sophisticated blade structural model exists that is able to account for blade torsion. Nonetheless, it was chosen to use ElastoDyn in this study for two reasons. The first reason is to speed up the OpenFAST calculations, as ElastoDyn requires less computational resources to run. The second reason is that by using a simpler structural model in OpenFAST, the impact of this choice on the global dynamics and loads of the chosen floating systems can be evaluated.

**3.3 Control**

In all three models the ROSCO v2.4.1 open-source controller (Abbas et al., 2022) is used. This controller has been selected as it is open-source and it includes an automatic tuning toolbox that can be used to determine the proportional and integral gains of the blade pitch controller in a simple manner (Lenfest et al., 2020). A traditional $K\omega^2$ law is used for the torque controller below rated wind speed. Above rated wind speed constant-torque control strategy is used. The pitch controller gains are tuned using ROSCO controller's automatic pitch-tuning routine based on the OpenFAST models of the two rotors. The controller includes a nacelle-velocity feedback loop developed especially for FOWTs, with the objective of avoiding negative blade-pitch controller damping that can occur in the case of FOWTs. However, this feature is not used in this study. The reason for this being that the feature did not work for the DeepLines models, as the required nacelle velocity sensor was not available as a controller input in

this code. In order to have a fair comparison between all codes, we decided to disable this feature and instead tuned the pitch controller to have lower PI-feedback terms. The natural frequencies and damping ratios of the pitch controller used for the three models are shown in Table 2. For all three models the natural frequency of the blade pitch controller is set below the platform pitch natural frequency, mitigating possible controller-driven system instabilities. Despite this, a certain degree of blade pitch-induced platform motion is noted, especially in the Softwind test-case, at near-rated wind speeds. The phenomenon impacts QBlade simulations more than OpenFAST and DeepLines simulations. The reason for this difference is probably linked to slight differences in the aerodynamic models that cause different controller reactions, as explained in detail in Sect. 4.3.1.

In the OC4 model, a peak-shaving minimum pitch saturation schedule is considered. Peak shaving is used to reduce loads near rated wind speed by imposing a minimum pitch angle as a function of the low pass filtered wind speed at hub height, as explained in (Abbas et al., 2022). In this model the same settings are used as in the public example that can be found in the ROSCO repository.

In DLC 1.4 shut-downs are performed by overriding the blade pitch controller with a specified pitch to feather maneuver in each code. The pitch to feather maneuver is initiated 5 seconds after the wind gust peak, as if the controller was reacting to the detection of an extreme yaw error and the blades are pitched at a speed of 10 °/s. In DeepLines the pitch to feather maneuver is longer in duration due to a setup difference. In fact, a specific pitch rate during a pitch to feather override maneuver cannot be specified in DeepLines, which needs a start and end time of the operation. Therefore, depending on the initial blade pitch angle, which depends on the coupled simulation and is thus different for each turbulent seed and each code, this can result in different pitch rates.

Table 2: Controller natural frequencies and damping ratios for the three test-cases.

| Model | Nat. f ($\omega$) | Damping ratio ($\beta$) |
|---|---|---|
| NREL 5MW OC4 | 0.2 [rad/s] | 1 [-] |
| DTU 10MW Softwind | 0.14 [rad/s] | 1 [-] |
| DTU 10MW Hexafloat | 0.114 [rad/s] | 1 [-] |

## 3.4 Hydrodynamics

For the OC4 and Softwind designs a Potential flow with Morison Drag (PFMD) approach is used in both OpenFAST and QBlade, whereby hydrodynamics are modeled by combining a potential flow solution with quadratic drag computed with Morison's equation (ME). Full difference-frequency Quadratic Transfer Functions (QTFs) are used in both QBlade and OpenFAST in the OC4 design. They were computed and provided for this geometry by ECN using NEMOH (Kurnia et al., 2022), a potential flow hydrodynamic solver developed by ECN. On the Softwind design, quadratic hydrodynamic excitation forces are included with Newman's approximation (Faltinsen, 1993). The same hydrodynamic coefficients are used for each design in all three models. Buoyancy is modeled differently in the three codes: QBlade and DeepLines model this force explicitly. The spar structure is divided into a series of cylindrical sections and buoyancy forces are discretely applied. OpenFAST on the other hand models buoyancy force as constant term and a linear stiffness matrix to include the contributions of buoyancy to the restoring forces on

the platform. Moreover, QBlade and DeepLines are able to model Wheeler wave stretching, which may introduce additional
non-linear forcing. In the Hexafloat model a different approach is used. In fact, the floater is made of relatively slender braces
that can be adequately modeled with a ME approach (Faltinsen, 1993). The same added mass and drag coefficients in both the
axial and transversal directions are used in DeepLines and QBlade, and the hydrodynamic forces predicted by the two codes
match well (Perez-Becker et al., 2022). The improvements implemented in QBlade to capture the slow-drift hydrodynamic forces
described in ((Behrens De Luna et al., 2023), Sect. 3.4), are not used in this study, and all three models share the same basic
hydrodynamic model, with the respective differences highlighted in this Section.

## 4 Results

In this Section the most relevant results are presented. General statistical information is presented first, followed by a selection
of ultimate loads recorded in DLCs 1.3 – 6.1 (Table 1) and a selection of lifetime DELs to compare fatigue load predictions. The
Softwind design is used as the design of choice in most cases as it features all three codes, and results from the other two designs
are also discussed when necessary. We were unable to complete all the simulations in all three codes in the comparison due to
numerical convergence issues. In particular, one out of sixteen simulations in DLC 6.2 in the Softwind model was not completed
in OpenFAST because of instabilities in the structural solver. Moreover, we were unable to complete all simulations in DLCs
1.2 (498/504), 1.3 (86/99), 6.1 (12/18), 6.2 (12/16) and 6.3 (12/18) in DeepLines. Similar issues are also present in the Hexafloat
model in DeepLines, where simulations did not converge in DLCs 1.2 (497/504), 6.1 (12/18), 6.2 (12/16) and 6.3 (12/18). The
cause of the incomplete runs can again be traced back to numerical instabilities in the solution. We chose not to attempt re-
running the simulations with a fine-tuning of the numerical solution scheme parameters because of budget and time constraints
within the project. Therefore, while not an inherent limitation of the code, this result is what could be achieved by a prepared
operator within the project timeline, which is also comparable to that of an industrial project. We were able to complete all the
simulations in QBlade. Results have shown good agreement between the codes in DLCs where the machine is operating, but
some discrepancies when the machine is parked. Moreover, generally larger differences in fatigue loads than in extreme loads
between the codes are noted.

## 4.1 Statistical Comparison

Figures 3 and 4 show a statistical comparison of selected operational sensors over the working range of the wind turbines. The
wind speed is extracted at 100 m above mean sea water level. The markers represent the mean values recorded in DLC 1.2, the
shaded area corresponds to twice the standard deviation of the signal for each wind speed and the dashed lines show the minimum
and maximum values recorded during the DLC 1.2 runs. Control sensors, often used to monitor the operation of the wind turbine,
are shown in Fig. 3. Although global trends are the same for all three codes in all three test-cases, some important differences
can be pointed out. With respect to QBlade, mean aerodynamic thrust is lower for DeepLines in the Softwind and Hexafloat test
cases at below rated wind speed and is also lower for OpenFAST in the OC4 test-case. In the case of the OC4 test-case, the
difference in thrust can, at least partially, be attributed to differences in rotor speed (Fig. 3 (h)). In fact, mean rotor speed is
higher in QBlade, causing the rotor to operate at a higher tip speed ratio (TSR), leading to a higher thrust coefficient. Similar
differences in this regard were noted also in previous comparisons between QBlade and OpenFAST (Perez-Becker et al., 2020).
For the Softwind and Hexafloat test-cases (Figs. 3 (b, e)), less difference in rotor speed can be noted, and the difference in thrust
is therefore more likely to be caused solely by differences in the aerodynamic models. The differences in aerodynamic modeling
are also apparent when analyzing blade pitch statistics in Figs. 3 (c, f, i). In fact, while good agreement in mean values can be
noted for QBlade and OpenFAST, mean blade pitch is lower for DeepLines through most of the wind speed range. In addition,
the difference between maximum and minimum blade pitch angles is larger for DeepLines with respect to OpenFAST and
QBlade. Moreover, as shown in Fig. 3 (b, e), minimum rotor speed is not enforced in DeepLines, and the rotor operates at lower
rpm at cut-in in both the Hexafloat and Softwind test cases. The ROSCO controller that was used in this code-to-code comparison
required recompiling to be used in DeepLines Wind because the blade pitch and twist angle conventions that are used in this
code differ from those used in QBlade and OpenFAST and as a result, minimum rotor speed is not enforced in DeepLines. To
the best of our knowledge, the controller is functionally identical to that used in OpenFAST and QBlade in all other aspects.
This influences fatigue loads, especially edgewise and in-plane blade root bending moments, that are strongly dependent on
cyclic gravitational loading. On the other hand, we can assume the influence of this discrepancy on extreme loads to be limited,
as these loads are recorded at higher mean wind speeds.

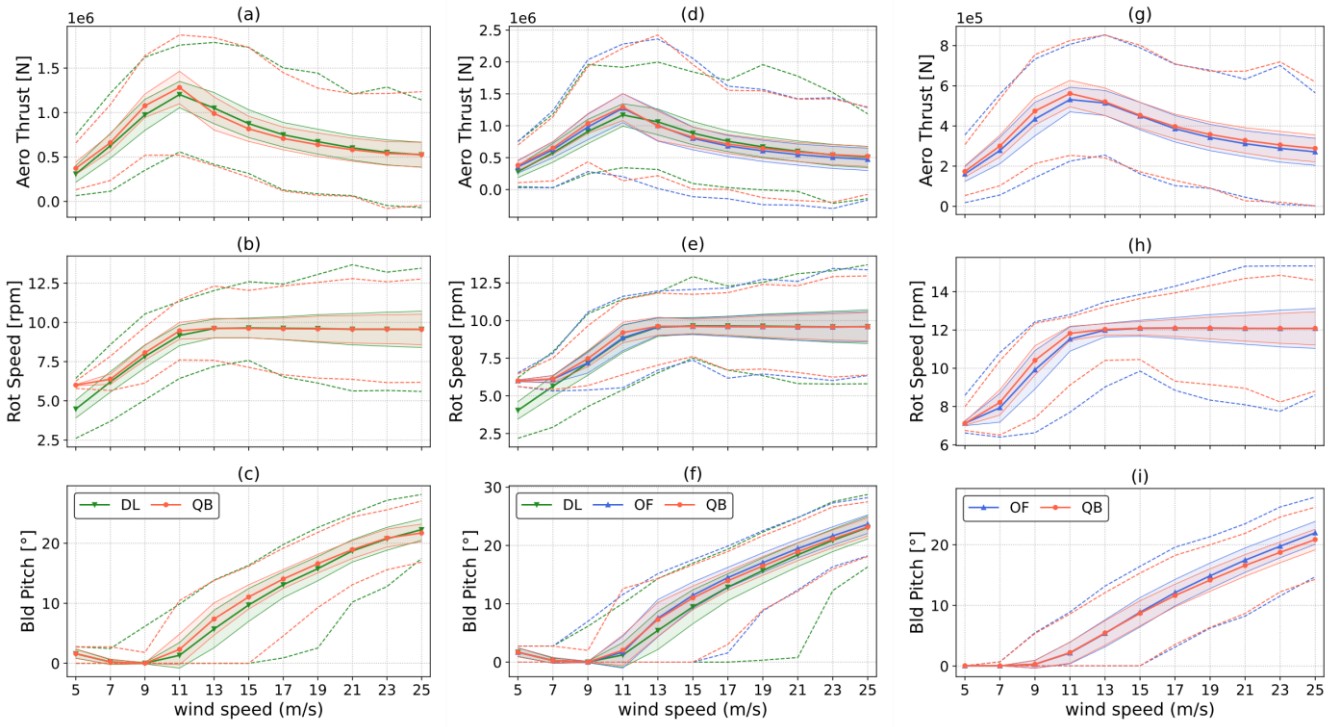

**Figure 3: Statistics of aerodynamic thrust (a, d, g), rotor speed (b, e, h) and blade pitch (c, f, i) as a function of mean wind speed**
**recorded in DLC 1.2. Solid lines with markers represent mean values, shaded areas represent twice the recorded standard deviation,**
**dashed lines for the minimum and maximum recorded values. DTU 10MW Hexafloat (a-c), DTU 10MW Softwind (d-f) and NREL**
**5MW OC4 (g-i).**
In Fig. 4, statistics of platform pitch and mooring line tension are shown. For the Softwind and Hexafloat test-case one of the
two upwind mooring lines is chosen, while for the OC4 test-case the tension of the upwind mooring line is reported in Fig. 4 (f).
As for the control sensors shown in Fig. 3, good general agreement can be seen for all three codes in all three test-cases. Platform
pitch is remarkably similar in mean value, standard deviation, and minimum/maximum value for the OC4 test-case (Fig. 4 (e)),
although higher standard deviation can be noted for wind speed near cut-in and cut-out in OpenFAST. This is interesting because
a higher platform-pitch standard deviation indicates increased gravitational and inertial loading variations on the tower. Very
good agreement between OpenFAST and QBlade is also shown in Fig. 4 (c). Despite platform pitch standard deviation being
lower in QBlade for most wind speeds, at 13 m/s mean wind speed it is higher for QBlade. A similar trend can also be noted in
Fig. 4 (a), where again the standard deviation of blade pitch is higher for QBlade at 11 m/s and 13 m/s mean wind speeds.
Analyzing the time series of the various codes at these wind speeds reveals that the increased standard deviation in QBlade near
rated is a result of blade pitch – platform pitch self-excitation. This phenomenon is discussed in detail in Sect. 4.3. Mooring line
tensions are in good agreement in all three test-cases although some differences can be noted. The largest difference is shown in
Fig. 4 (d), where a significant difference in mean tension can be noted between DeepLines and the other codes. Such difference
is a result of different model tuning, as discussed in Sect. 2.3.2.

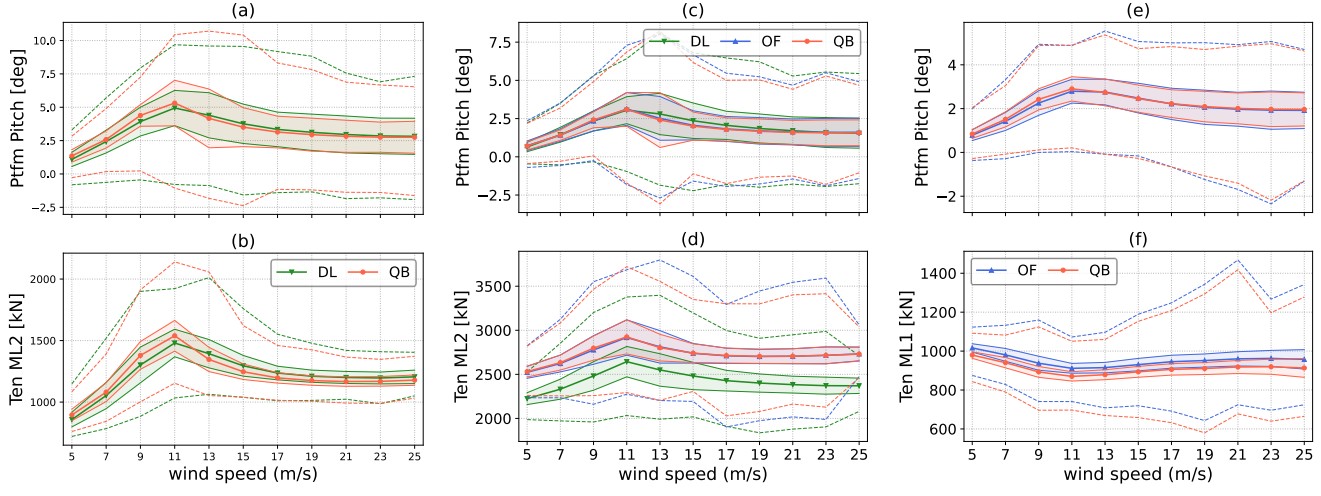

**Figure 4: Statistics of platform pitch (a, c, e), upwind mooring line tension (b, f) and tendon tension (d) as a function of mean wind**
**speed recorded in DLC 1.2. Solid lines with markers represent mean values, shaded areas represent twice the recorded standard**
**deviation, dashed lines for the minimum and maximum recorded values. DTU 10MW Hexafloat (a-b), DTU 10MW Softwind (c-d) and**
**NREL 5MW OC4 (e-f).**
**4.2 Ultimate Loads**
This Section presents the ultimate loads, computed with the maximum averaging method described in Sect. 2.4, for key selected
load sensors. This Section is focused on understanding which phenomena and modeling differences may influence the prediction

of extreme loads. The analysis focuses on maximum extreme loads only, disregarding minimum loads to streamline the discussion. Minimum extreme loads are reported in Appendix A. In Fig. 5, the ratios of selected ultimate loads on the turbine with respect to the values obtained in QBlade, assumed here as benchmark, are shown. The DLCs in which the respective maximums are recorded are also reported for each of the bars in Fig. 5. For blade root bending moments, the maximum value recorded across the three blades is shown. Figure 5 also reports the blade where the peak load is recorded. Ultimate loads are recorded across all the DLCs, thus encompassing both power production and parked load cases, depending on the specific load sensor and FOWT design being examined. In the OC4 test case (Fig. 5 (c)) extreme loads are predicted in the same DLC in OpenFAST and QBlade, with the exception of blade root in-plane bending moment (BR Mxc). This FOWT design is the one where the best overall agreement between the compared codes was reached. In the Softwind and Hexafloat designs, extreme loads are recorded in different DLCs for some load sensors, as is the case for TT Fx for Softwind and BR Myc for Hexafloat. In both cases extreme loads predicted across multiple DLCs are very close in magnitude, causing the ultimate extreme load to be predicted in different DLCs depending on the specific model's response.

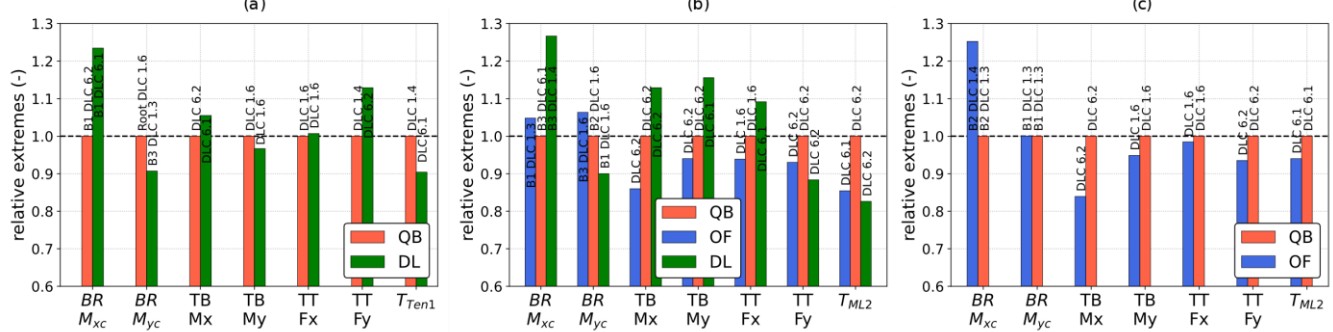

**Figure 5: Selection of ultimate loads (maximum) recorded in the three simulation codes. (a) DTU 10MW Hexafloat, (b) DTU 10MW Softwind and (c) NREL 5MW OC4.**

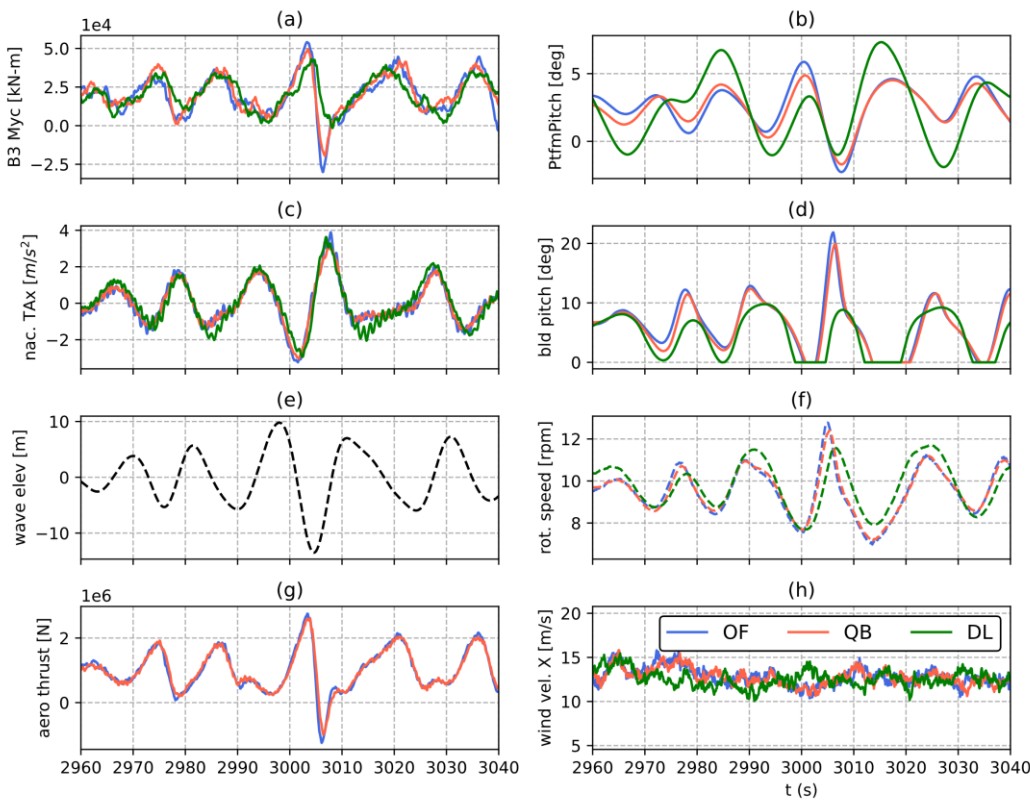

**Figure 6: Time series of out-of-plane root bending moment of blade 3 of the Softwind model in DLC 1.6, (ws = 11 m/s, Hs = 9), where**
**maximum bending moment is recorded for OpenFAST. From top to bottom: B#3 out-of-plane root bending moment (a), platform**
**pitch (b), nacelle fore-aft acceleration (c), blade pitch (d), and wave height at platform reference position (e), rotor speed (f),**
**aerodynamic thrust (not available in DeepLines outputs) (g), wind speed at hub height (h).**
**4.2.1 Blade Root Extreme Loads**
Regarding blade root bending moments, there is larger variation in BR Mxc ultimate load than BR Myc. BR Myc is much higher
in magnitude than BR Mxc and thus has a greater influence on component design. Nonetheless, BR Mxc is approximately 23%
higher on the Hexafloat test-case for DeepLines, and 27% higher in the Softwind test-case. Similarly, BR Mxc is approximately
25% higher for OpenFAST in OC4. Out-of-plane blade root bending moments are in better agreement, DeepLines predicting
10% lower loads than QBlade in the Hexafloat and Softwind test-cases, while OpenFAST and QBlade are much closer, the
former being 5% higher in Softwind and nearly identical to QBlade in OC4.
The out-of-plane blade root bending moments are mostly influenced by aerodynamic loading, as lift force is directed mostly out-
of-plane. On a FOWT however, the coupled dynamics of the entire system influence these load sensors. This is demonstrated in
Fig. 6, where the time series of multiple load sensors, including BR Myc, platform pitch, aerodynamic thrust and nacelle fore-
aft acceleration are shown at the time instant where the maximum BR Myc in OpenFAST is recorded. When the load peak is
recorded the wind speed is rising and is around the rated wind speed value. In addition, an extreme wave impacts the substructure.

The latter causes the FOWT to move, as shown in the platform pitch and nacelle fore-aft acceleration sensors time series. In turn this causes large relative inflow variations on the rotor. As hydrodynamic forces cause the platform to swing forward, rotor thrust increases causing BR Myc to peak. Due to the increase in relative inflow, rotor speed increases (Fig. 6 (d)) and the controller reacts by aggressively pitching the blades, especially in QBlade and OpenFAST. While controller response depends on and influences the global response of the system, one reason for the different controller reactions in DeepLines is the different wind speed in this code (Fig. 6 (e)). In fact, the same wind fields are used in all three codes, but a time-shift is present in DeepLines with respect to the other models due to differences in how the wind fields are imported. In fact, depending on the simulation tool, wind fields are often shifted on import in order to make sure that the turbine is fully immersed in the wind field in case of yaw misalignment. On the other hand, no such shift is present in the wave fields. Therefore, environmental inputs are out of sync if OpenFAST and QBlade are compared to DeepLines. The increase in blade pitch is able to limit rotor speed overshoot but causes a sudden decrease in rotor loading, which in turn is the cause of BR Myc reaching a local minimum shortly after peaking. Therefore, platform motion influences BR Myc indirectly: not through variation in inertial and gravitational loads but through variation in aerodynamic loading. In summary, even small differences in aspects such as input conditions, hydrodynamics, aerodynamics, control, and overall set-up definition can influence ultimate loads through different system dynamic behavior.

### 4.2.2 Tower Base Extreme Loads

Shifting focus to tower base loads, fore-aft (TB My) are, similarly to blade root loads, greater in magnitude than side-side loads (TB Mx) that will thus be treated briefly. Side-side tower base bending moment (TB Mx) ultimate load always occurs in parked conditions for all three test-cases and all three design codes. Moreover, except for DeepLines in the Hexafloat test-case, ultimate loads always occur in DLC 6.2, where in addition to +/- 30° incoming wave heading, yaw misalignment is present.

For brevity reasons, we chose to focus mostly on fore-aft loads in this study, which are higher than in-plane/side-side loads, as the incoming wind is always directed in the fore-aft direction in this study. However, as shown in further detail in (Papi et al., 2023), in all three test-cases a strong correlation between platform roll and side-side tower base bending moment (TB Mx) is present, indicating that these ultimate loads are hydrodynamics-driven. In fact, as the RNA and tower are heavy components, gravitational and inertial loads can be significant on FOWT towers. Regarding specific test-cases, in OC4 TB Mx ultimate load is approximately 16% lower in OpenFAST. This discrepancy is mainly caused by response at the tower natural frequency in QBlade, which is not present in OpenFAST. On the other hand, if time series of TB Mx are compared for the Softwind test-case, little variation can be noted between the three codes. For this load sensor the difference between QBlade and OpenFAST ultimate loads that is shown in Fig. 5, is amplified by the maximum averaging technique. As described in Sect. 2.4, the ultimate load in load cases with multiple turbulent seeds is computed as the maximum value closest to the mean of the maximums recorded across all the turbulent seeds. Therefore, because ultimate loads are slightly different in QBlade and OpenFAST, the peak load closest to the mean is recorded in different seeds for the two codes. This demonstrates how small differences between the models can be amplified by the post-processing technique.

Maximum tower base fore-aft bending moment (TB My) is also recorded in parked conditions in the Softwind test-case - DLC 6.2 for QBlade and OpenFAST and DLC 6.1 for DeepLines. Analyzing the times series of TB My in DLC 6.1 (Fig. 7) when peak load is recorded in DeepLines, the ultimate load is generated by a combination of gravitational and inertial loading resulting from platform motion. Higher values of platform pitch are noted in DeepLines, possibly a result of the slacker mooring lines in DeepLines, which explain the higher TB My. On the other hand, in the Hexafloat and OC4 test-cases, maximum TB My is found in DLC1.6 for all codes (Fig. 5). In both the latter cases OpenFAST and DeepLines are approximately 5% and 3% lower than QBlade in this metric. In this case ultimate loads are recorded around rated wind speed, similarly to BR Myc. Differently from the latter, which is analyzed in detail in Fig. 6, in the case of TB My, platform motion contributes directly to tower base loading as it increases gravitational and inertial forces. Overall, the three codes are close in this metric confirming that all three are able to capture the system dynamics in presence of extreme waves to a similar degree.

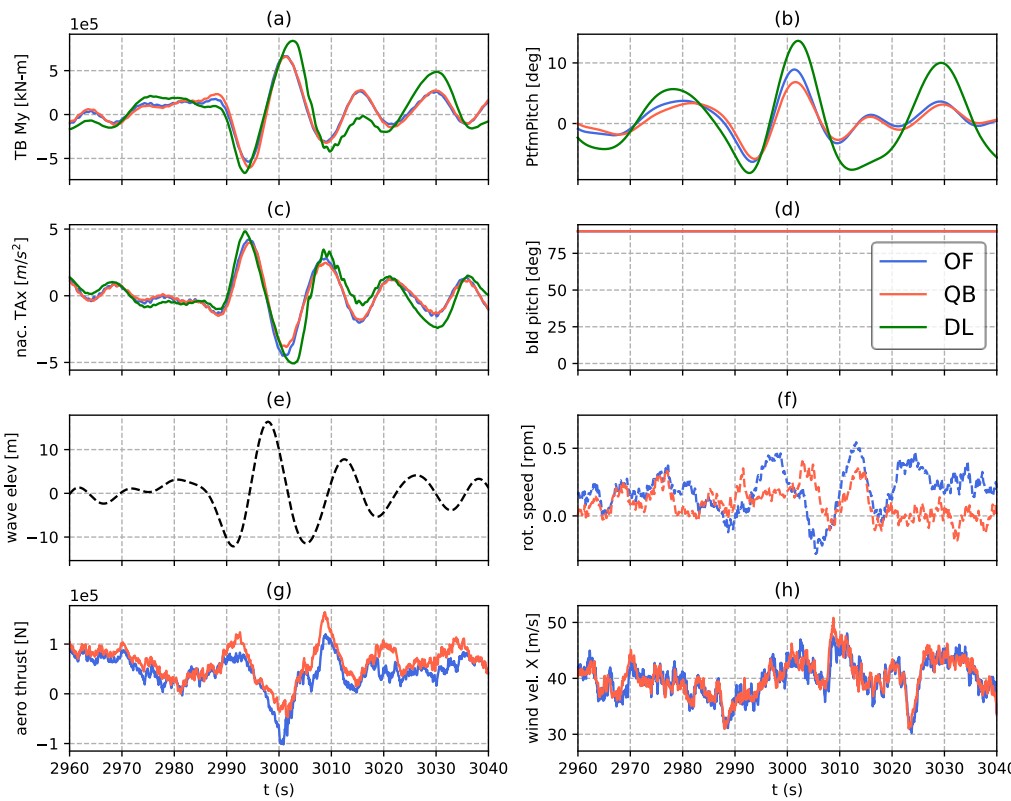

**Figure 7: Time series of fore-aft tower base bending moment of the Softwind model in DLC 6.1, (ws = 37 m/s, Hs = 16.5), where maximum bending moment is recorded for OpenFAST. Tower base fore-aft bending moment (a), platform pitch (b), nacelle fore-aft acceleration (c), blade pitch (d), and wave height at platform reference position (e), rotor speed (f), aerodynamic thrust (not available in DeepLines outputs) (g), wind speed at hub height (h). DeepLines data are missing in (d,f,h) as these data cannot be exported from the code when the controller is not used.**

### 4.3 Fatigue Loads

#### 4.3.1 Blade Root Fatigue Loads

Lifetime, zero-mean DELs computed with the procedure highlighted in Sect. 2.4 at blade root in the coned coordinate system are shown in Fig. 8. Contrary to extreme loads, a clear trend is apparent in this case. In fact, with respect to QBlade, Lifetime DELs are lower in DeepLines but higher in OpenFAST. In particular, 1Hz DELs are 3-5% lower than QBlade for DeepLines, with little variation across the three blades. Indeed, fatigue loads are consistent among the three blades for all three codes and all three test-cases, indicating good statistical convergence. Comparing QBlade and OpenFAST, blade root fatigue loads are very close (0-3%) in case of the OC4 test-case, while increases of up to 12% in out-of-plane blade root bending moments can be seen for Softwind. On the other hand, OpenFAST and QBlade are closer in the prediction of in-plane root bending moments than out-of-plane root bending moments. The former are mainly driven by gravity, explaining the smaller differences between the compared wind turbine simulation codes.

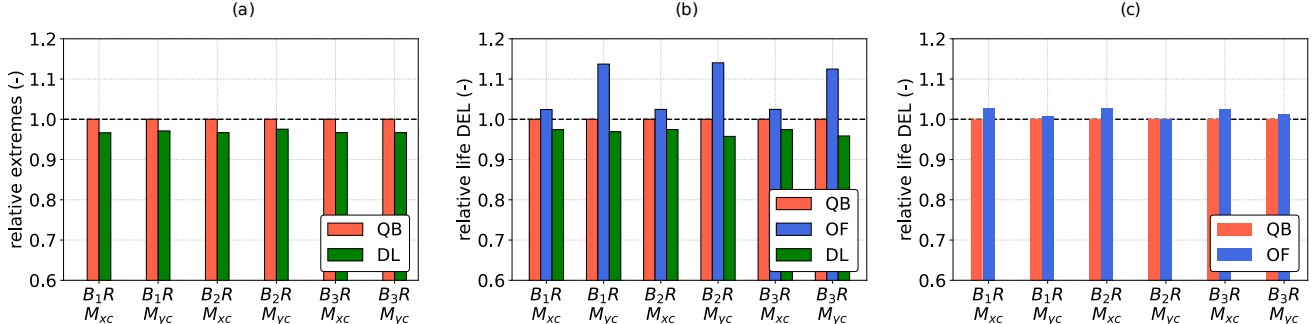

**Figure 8: Blade root fatigue loads in coned coordinate system: lifetime DELs normalized respect to values computed in QBlade. From left to right: DTU 10MW Hexafloat, DTU 10MW Softwind and NREL 5MW OC4.**

To better understand the differences in Lifetime DELs, the Cumulative Power Spectral Density (CPSD) of blade root bending moments for the Softwind FOWT design are shown in Fig. 9. They are obtained as the cumulative sum of the PSD of the signal. A CPSD plot is read from left to right; steps in the data indicate peaks in the underlying PSD. When comparing two signals, the increase or decrease in distance between the lines indicates the differences between them. The CPSDs for the Hexafloat FOWT design look very similar and are not shown here for brevity as similar conclusions can be drawn. At all three of the examined wind speeds (7 m/s, 13 m/s and 23 m/s) 1P loads are the main contributors to in-plane fatigue loading (BR Mxc). The magnitude of 1P excitation is lower in DeepLines for all three wind speeds. The most relevant differences in this regard can be seen at 7 m/s (Fig. 9 (a)) and can be explained by the difference in rotor speed that was noted in Fig. 3. Because minimum rotor speed is not imposed in DeepLines, while it is in QBlade and OpenFAST, the 1P peak spans a larger frequency range in the former and is lower in magnitude.

Differences are also present in the BR Myc CPSD. The near absence of response between 1P and 2P, at wave frequency, indicates that apparent wind variations caused by platform motions do not induce relevant fatigue loading for this FOWT design. Three

distinct phenomena drive the differences in this load sensor at the three wind speeds shown in Fig. 9. At 7 m/s (Fig. 9 (d)) wind speed OpenFAST and DeepLines show higher low-frequency excitation than QBlade. This phenomenon deserves further attention and will be discussed later in this Section when similar results for the OC4 FOWT design are presented. Moreover, while small in magnitude when compared to low-frequency response, the 1P peak is larger in OpenFAST. 1P BR Myc load variation remains larger for OpenFAST across the wind speed range, but are most noticeable at 23 m/s (Figs. 10 (f))).

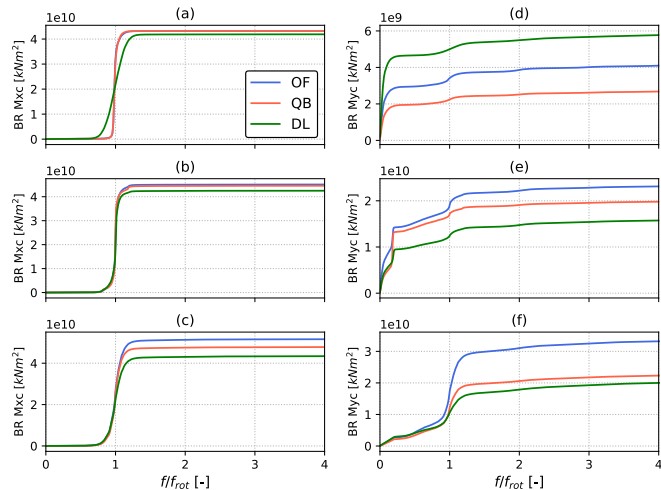

**Figure 9: Cumulative Power Spectral Density (PSD) of blade root in-plane (a-c) and out-of-plane (d-f) bending moment for the Softwind test-case. Frequency is normalized by mean revolution frequency. PSD is computed on all simulations with 7 m/s (a, d), 13 m/s (b, e) and 23 m/s (c, f) mean wind speed.**

Finally, at 13 m/s the three codes differ mainly in the low-frequency region, where the predicted response in OpenFAST is larger. Moreover, at this wind speed a large peak at the floater pitch natural frequency can also be seen, especially for QBlade. This peak in response at the floater natural frequency is caused by blade pitch – floater pitch self-excitation. As described in detail in (Larsen and Hanson, 2007), on a FOWT an increase in blade pitch causes aerodynamic loads to decrease, and the platform to swing forward as a consequence. In turn this causes the apparent wind speed on the rotor to increase and rotor speed to follow. The controller will thus react to the increased rotor speed by increasing blade pitch even further. A similar unstable behavior is triggered by a decrease in blade pitch, in this case the platform swings backward, reducing apparent wind speed and rotor speed, promoting further blade pitch reductions. As explained in Sect. 3.3, controller gains were reduced to avoid this phenomenon (see (Larsen and Hanson, 2007) for a detailed explanation on the effectiveness of this strategy). Despite this, as confirmed by the increased platform pitch standard deviation in Fig. 3 and blade pitch standard deviation in Fig. 4, unstable behavior emerged at 11 and 13 m/s wind speed. This can be seen clearly in Fig. 10, where the time series of platform pitch and blade pitch for the three FOWT designs during a 13 m/s DLC 1.2 simulation are shown - and also in Fig 17 (d) later on in this study. In Fig. 10, the OC4 model is not affected by pitch self-excitation, while the Hexafloat and Softwind models are. In the latter two models, DeepLines is the least influenced by the phenomenon and QBlade is the most affected, despite all three codes using the same controller.

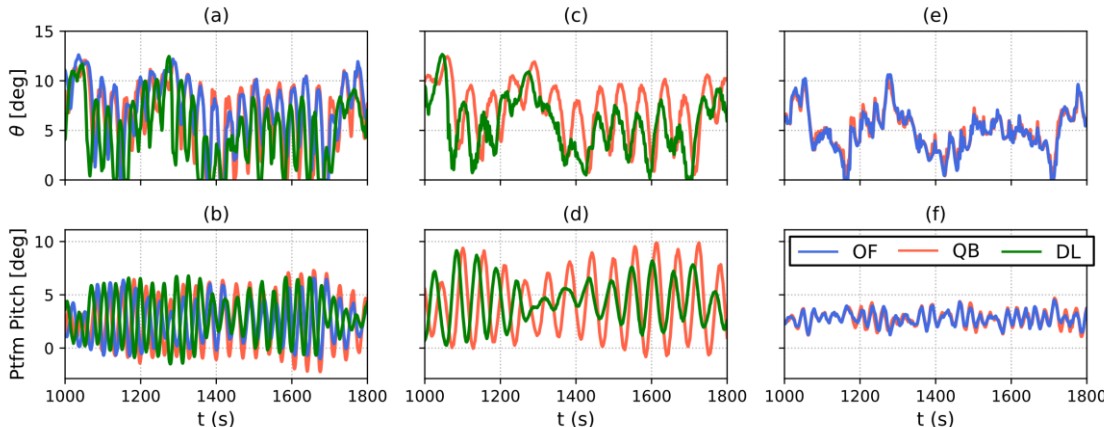

**Figure 10: Time series of blade pitch (top row) and platform pitch (bottom row) for a 13 m/s simulation in DLC 1.2. Softwind (a, b), Hexafloat (c, d) and OC4 (e, f).**

Various physical phenomena could cause such a difference in excitation. However, by process of exclusion, differences in hydrodynamic excitation are unlikely to be the cause of the increased self-excitation in QBlade, as nearly identical response in QBlade and OpenFAST was noted at the Softwind's pitch natural frequency in part one of this study ((Behrens De Luna et al., 2023), Fig. 13). Moreover, the way unsteady aerodynamics are modelled is also not the cause, as switching to DBEM in QBlade did not improve agreement in this regard with respect to OpenFAST (not shown herein for brevity). In addition, as stated previously, OpenFAST does not include blade torsion. However, switching to a rigid structure did not improve the agreement of OpenFAST and QBlade. A possible explanation for the difference in blade pitch - platform pitch self-excitation was put forward in part one of this study (Behrens De Luna et al., 2023) and is related to increased aerodynamic torque variation in QBlade with respect to the other two codes. Indeed, upon further investigations, differences in the system dynamics, and how they interact with the control system, could explain the observed behavior. As explained in detail by Abbas et al., (2022), the controller and turbine can be seen as a closed-loop second-order system, characterized by a natural frequency at a certain operating wind speed:

$$\omega^2 = k_i(U_{op})B = k_i(U_{op})\frac{N_g}{J}\frac{\partial \tau_a}{\partial \beta} \tag{1}$$

where $N_g$ and $J$ are the gearbox ratio and rotor inertia, which are the same in OpenFAST, QBlade and DeepLines. The higher the natural frequency, the more responsive the system is to an external disturbance such as a platform pitch oscillation. The integral controller gain $k_i$ is also the same in the two codes, as it depends on the controller tuning. The slope of the aerodynamic torque as a function of blade pitch is, however, different in the two codes. The derivative of aerodynamic torque as a function of blade pitch for the mean 11 m/s operating conditions is shown in Fig. 11 (b). As $\frac{\partial \tau_a}{\partial \beta}$ is larger in magnitude for QBlade at the mean operating blade pitch of approximately 0.5°, from eq. 1, $\omega^2$ is also larger, leading to increased self-excitation in

QBlade. This highlights how small differences in aerodynamics can lead to different controller response and influence turbine
load predictions significantly.

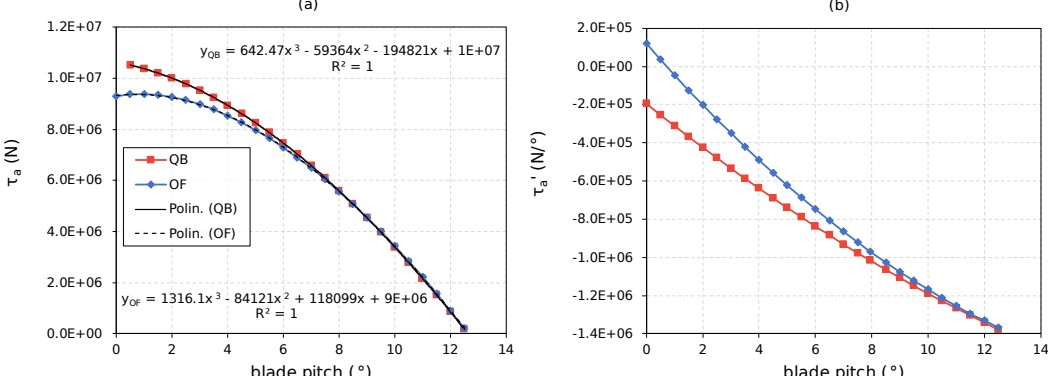

**Figure 11: (a) aerodynamic torque as a function of blade pitch for OpenFAST and QBlade for 11 m/s operating TSR,**
**and relative trendlines. (b) derivative of aerodynamic torque as a function of blade pitch computed from analytic**
**derivative of trendlines.**
Despite QBlade and OpenFAST lifetime DELs being very close, the OC4 FOWT design highlights some interesting behavior,
and differs in some key aspects from the Softwind FOWT design. CPSDs of blade root bending moments can, again, help
investigate the causes of the differences in Lifetime DELs and are shown in Figure 12. Focusing on out-of-plane root bending
moment (TB My), differences in 1P excitation that are highlighted for the Softwind design (Fig. 9) are not apparent in OC4. The
larger difference in 1P excitation between models on the Softwind design with respect to the OC4 design can likely be explained
by the size difference of the two rotors. As found by Madsen et al., (2020) non-uniform rotor loading due to turbulence and wind
shear increases with rotor size. For a larger rotor, a higher portion of the turbulent flow structures feature a length scale that is
smaller than the rotor diameter, shifting a higher ratio of the total energy in the turbulent spectrum from lower frequencies to the
1P frequency and multiples. As for wind shear, a larger rotor operates in a larger portion of the atmospheric boundary layer,
meaning that each blade experiences more inflow variation during a revolution. As these phenomena increase in magnitude they
are expected to increase the differences between aerodynamic models at 1P frequency.

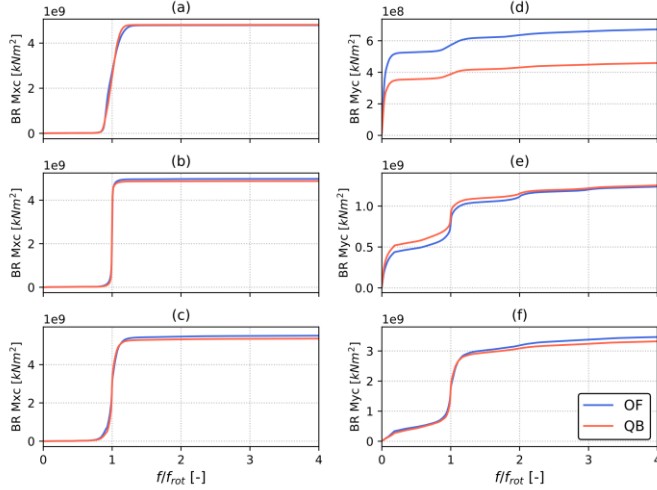


**Figure 12: Cumulative Power Spectral Density (CPSD) of blade root in-plane (a-c) and out-of-plane (d-f) bending moment for the OC4 model. PSD is computed on all simulations with 7 m/s (a, d), 13 m/s (b, e) and 23 m/s (c, f) mean wind speed.**

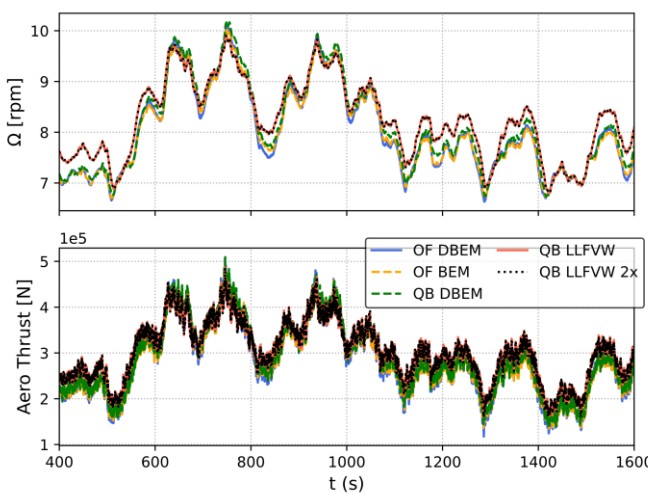


**Figure 13: Time series of rotor speed and aerodynamic thrust in a 7 m/s simulation of the OC4 test-case. Various wake models are compared; OpenFAST DBEM (Branlard et al., 2022), OpenFAST BEM (Ning et al., 2015), QBlade DBEM (Madsen et al., 2020) and QBlade LLFVW (Marten, 2020).**

On the other hand, the low frequency excitation difference that was noted for the Softwind design is also found for the OC4 design (Fig. 12 (d)) and, although not shown herein for brevity, is also found to be one of the main drivers of the higher Lifetime DELs in OpenFAST (Fig. 8). To better understand this difference, additional simulations were carried out with additional aerodynamic models in both QBlade and OpenFAST in an attempt to isolate the cause of such differences. In particular, OpenFAST simulations were performed using quasi-steady BEM without dynamic induction corrections (OpenFAST BEM). QBlade on the other hand was run using LLFVW with doubled wake length (LLFVW x2) and with the polar-BEM method (Madsen et al., 2020) (QBlade DBEM). Time series of rotor speed and aerodynamic thrust are shown in Fig. 13 for a 7 m/s mean

wind speed simulation in DLC 1.2. As shown in Fig. 13, larger variations in rotor speed can be noted in the BEM-based models. This phenomenon is present in both QBlade and OpenFAST and no improvement with respect to QBlade LLFVW is noted when a dynamic induction correction is used. On the other hand, doubling the wake length in the LLFVW simulation has little to no effect on rotor speed, indicating that the wake cut-off length used in the study is adequate. The larger rotor speed variation in BEM models causes rotor thrust to vary more as TSR varies, thus causing the additional low-frequency loading shown in Fig. 13.

These results can be put into perspective by comparing them to other authors' findings. Indeed, differences between BEM-based and LLFVW aerodynamic models in the prediction of blade root fatigue loads have also been noted by other authors. Boorsma et al. (2020) attributed the differences observed at 1P frequency to different induction tracking of the BEM models during blade revolution, which causes differences in aerodynamic loading amplitude if wind shear, yaw misalignment, rotor tilt and, in the case of FOWTs, platform pitch are present. In addition to 1P differences, Perez-Becker et al. (2020) also noted differences between LLFVW and BEM at low frequencies, the latter mainly being caused by different blade pitch actuation in the models. In the context of FOWTs, Corniglion (2022) also found blade root fatigue loads predicted with a LLFVW model to be lower than those computed with a BEM-based aerodynamic tool. In this context, the higher fatigue loads that are noted in OpenFAST are in line with these findings. However, the same cannot be said for DeepLines that predicts lower lifetime DELs than the LLFVW-based QBlade.

### 4.3.1 Tower Base and Mooring Fatigue Loads

Tower top, tower base and mooring lifetime DELs are shown in Fig. 14 for the three FOWT designs. The OC4 and Hexafloat designs show a similar trend to those shown in Fig. 8; lower lifetime DELs for DeepLines and higher Lifetime DELs for OpenFAST. However, the differences in fore-aft tower lifetime DELs (TT Fx, TB My) in Fig. 14 (a) and (c) are larger than those in out-of-plane blade root lifetime DELs in Fig. 8 (a) and (c). Such phenomenon can be traced back, at least in part, to the differences in platform pitch that are noted in Fig. 4, which cause larger or smaller variations in gravitational and inertial forces on the tower, increasing the difference in tower lifetime DELs. Differently from blade root fatigue loads however, OpenFAST and DeepLines show good agreement in terms of lifetime DELs in Fig. 14 for the Softwind design. Tower-related fatigue loads are lower than QBlade, while mooring line fatigue predictions are higher. Moreover, differences in side-side tower loads (TT Fy and TB Mx) appear to be smaller than those found in the respective fore-aft sensors (TT Fx and TB My). These load sensors are arguably less influenced by aerodynamics, as the wind is always aligned with the global X direction, and more influenced by hydrodynamics, as wave headings range from -150° to 150°. In this context the good agreement in side-side loads is expected as hydrodynamics are modeled similarly in all three codes.

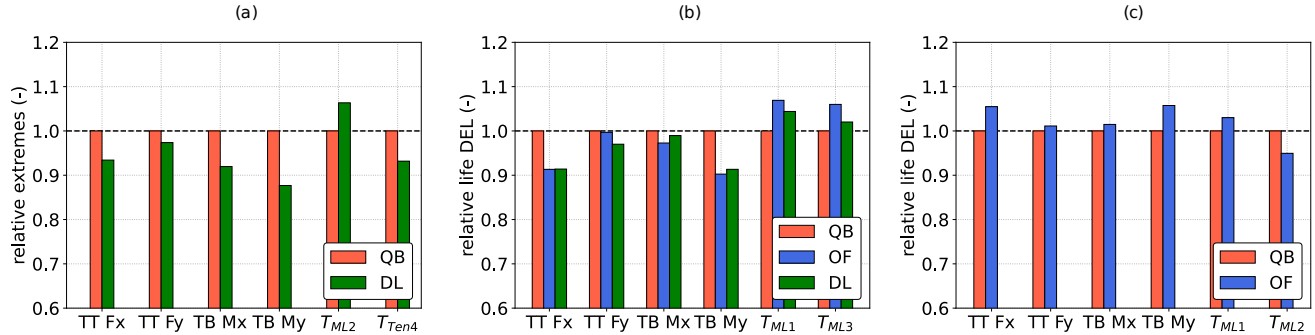

**Figure 14: Lifetime DELs normalized with respect to values computed in QBlade. Yaw bearing shear forces in p coordinate system and tower base fore-aft and side-side bending moments and shear forces in t coordinate system. From left to right: DTU 10MW Hexafloat, DTU 10MW Softwind and NREL 5MW OC4.**

The differences between the three models can be analyzed in more detail by comparing 1Hz DELs weighted by the probability of each environmental condition to occur:

$$\overline{DEL_i} = p_i * DEL = p_i \left( \frac{\Sigma_j \ n_j A_j{}^m}{t} \right)^{1/m} \tag{2}$$

$p_i$ is the probability of each condition to occur, $n_j$ and $A_j$ are the combinations of rainflow counted j-th number of cycles and amplitude in each simulation and m is the Wöhler curve exponent, equal to 10 for the composite blades and 4 for the other steel components. As discussed in Sect. 2.4, 1Hz DELs multiplied by their respective probability of occurrence are representative of the contribution to lifetime fatigue loads of each operating condition.

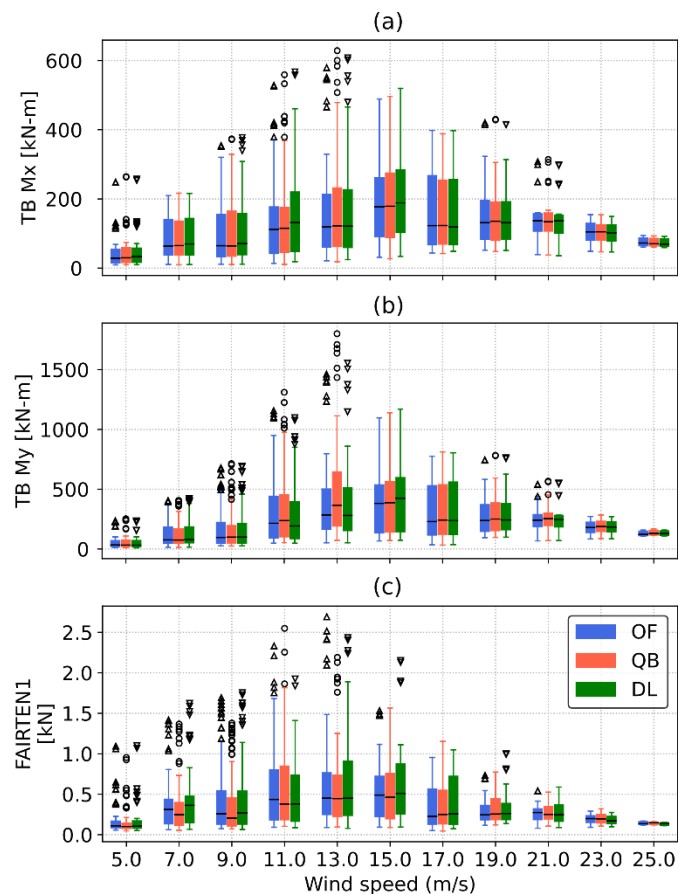


**Figure 15: Statistics of tower base bending moment and fairlead tension 1Hz zero-mean Damage Equivalent Loads weighted by the**
**probability of each environmental bin they refer to for the Softwind model. The boxes represent the 1ˢᵗ and 3ʳᵈ quartiles, the whiskers**
**represent the data range and are found by adding/subtracting to the box edges 1.5 times the interquartile (IQR) range, the horizontal**
**line is the median of the data and outlier values are shown as scatter points.**
Statistics of tower base and fairlead tension of one of the upwind mooring lines 1Hz DELs for the Softwind design are shown in
Fig. 15. From a fatigue damage standpoint, the most relevant wind speeds are included between 9 m/s and 19 m/s wind speed.
While 1Hz DELs are very close for all three numerical codes in Fig. 15 (a), the analysis of Fig. 15 (b) can help pinpoint the root
cause of the increased Lifetime DEL prediction in QBlade. In fact, while the three codes agree well across most wind speeds,
1Hz DELs are statistically higher for QBlade particularly in the 11 m/s and 13 m/s wind speed bins. The CPSDs of tower base
bending moments for the 7 m/s, 13 m/s and 23 m/s wind speed bins are shown in Figure 16. It stands out that tower base excitation
is dominated by low-frequency peaks, corresponding to the floater's natural surge/sway and pitch/roll natural frequencies, and
by response in the wave excitation frequency band. Moreover, contrary to blade root loads, 1P and 3P excitation is nearly
irrelevant as the CPSDs show a flat profile from 0.2 Hz and upwards.

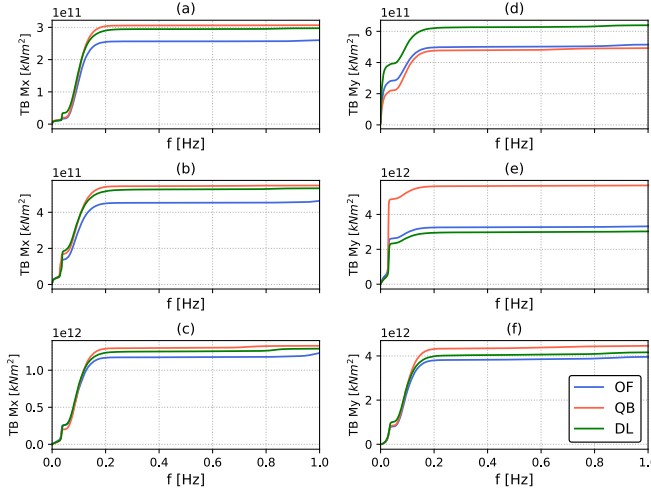

**Figure 16: Cumulative Power Spectral Density (CPSD) of tower base side-side (a-c) and fore-aft (d-f) bending moment for the Softwind test-case. CPSD is computed on all simulations with 7 m/s (a, d), 13 m/s (b, e) and 23 m/s (c, f) mean wind speed.**

Regarding fore-aft bending moment (TB My), at 7 m/s (Fig. 16 (d)), low-frequency aerodynamic excitation is the main driver of differences between QBlade – that shows lower response and fatigue loads at this wind speed – and the BEM-based codes. These differences are caused by the higher rotor speed variations recorded in OpenFAST and especially in DeepLines, as minimum rotor speed is not enforced in this code. The higher rotor speed variation leads to higher variation in aerodynamic forcing, as shown in Fig. 13. This phenomenon also contributes to the higher platform pitch variation that is observed for the BEM based codes (Fig. 4), further increasing low-frequency TB My excitation.

When analyzing Fig 16 (e), higher response at the floater pitch natural frequency is noted in QBlade. The cause of the increased response is floater-pitch blade-pitch instability, discussed in detail in Sect. 4.3.1

The same phenomenon also impacts the OC4 testcase, as shown in Fig. 17. The largest differences between OpenFAST and QBlade in the fore-aft tower base bending moment 1Hz DELs are located in the 9 m/s wind speed bin (Fig. 17 (a)). The CPSDs of aerodynamic thrust, platform pitch and TB My (Figs. 17 (b,c,d)) show that the main differences between the codes are found at very low frequencies, and are again caused by differences in aerodynamic response that are amplified by platform pitch and rotor speed variations.

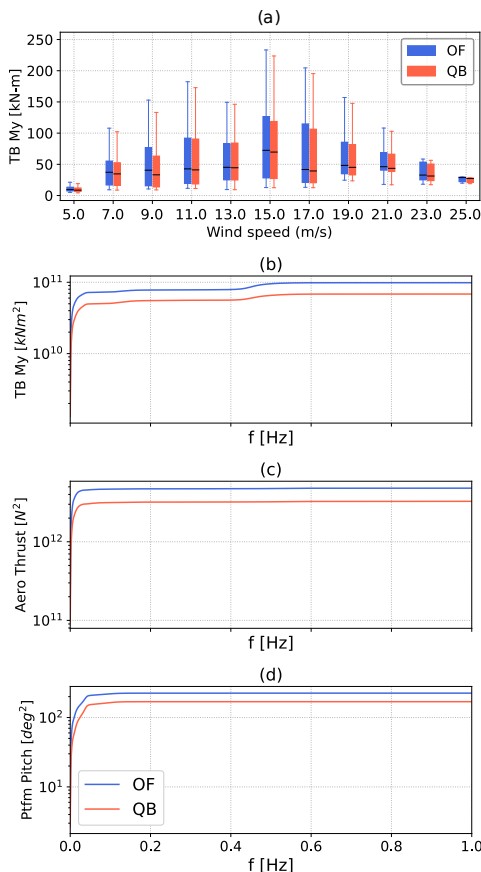

**Figure 17: (a) Statistics of tower base fore-aft bending moment 1Hz zero-mean Damage Equivalent Loads weighted by the probability of each environmental bin they refer to for the OC4 model. The boxes represent the 1st and 3rd quartiles, the whiskers represent the data range and are found by adding/subtracting to the box edges 1.5 times the interquartile (IQR) range, the horizontal line is the median of the data and flier values are shown as scatter points. (b,c,d) Cumulative Power Spectral Density (CPSD) of tower base fore-aft bending moment, aerodynamic thrust and platform pitch for the OC4 design. PSD is computed on all simulations with 9 m/s mean wind speed.**

Going back to the Softwind FOWT concept, at 13 m/s (Fig. 16 (e)) the largest difference between QBlade and the other codes is at the floater pitch natural frequency, where TB My PSD is much larger in the former code. The higher response is caused by the same phenomenon that causes higher blade root CPSDs at 13 m/s wind speed in QBlade (Fig. 9): floater and blade pitch self-excitation. In the case of tower base loads, in addition to cyclic variation in aerodynamic loads, cyclic inertial and gravitational forcing become relevant load sources, as the weight of the tower itself and the Rotor Nacelle Assembly (RNA) are considerable. Therefore, despite QBlade comparing well to the other two codes at other wind speeds (Fig. 16 (f)), the difference highlighted at 13 m/s (Fig. 16 (e)) ultimately leads to higher TB My lifetime DELs for QBlade (Fig. 14).

As shown in Fig. 18, floater and blade pitch self-excitation also influence fatigue load predictions for the Hexafloat model. As discussed previously, DeepLines predicts lower lifetime DELs than QBlade for this test-case. Contrary to floater-pitch-frequency

excitation, the peak in TB My response in correspondence of the tower first fore-aft natural frequency located at 0.2 Hz is
captured well by both DeepLines and QBlade (Fig. 18 (b,c)).

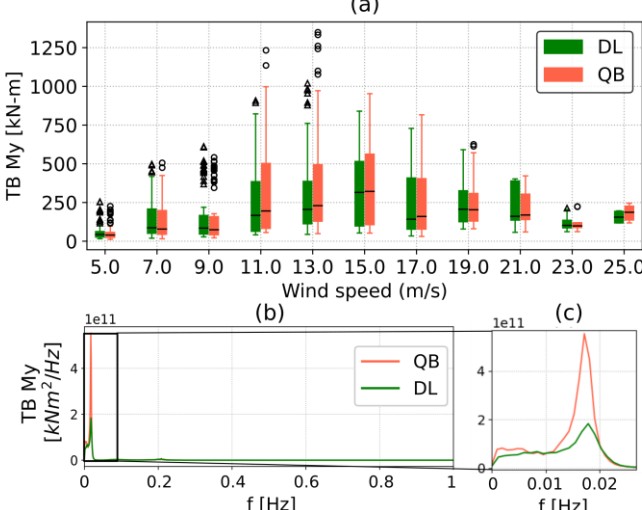

**Figure 18: (a) Statistics of fore-aft tower base bending moment 1Hz zero-mean Damage Equivalent Loads weighted by the probability**
**of each environmental bin they refer to for the Hexafloat model. The boxes represent the 1st and 3rd quartiles, the whiskers represent**
**the data range and are found by adding/subtracting to the box edges 1.5 times the interquartile (IQR) range, the horizontal line is the**
**median of the data and outlier values are shown as scatter points. (b) Power Spectral Density (PSD) of tower base fore-aft bending**
**moment for the Hexafloat test-case. PSD is computed on all simulations with 11 m/s mean wind speed.**

## 5 Conclusions

An extensive code-to-code comparison with realistic environmental conditions is performed in this study. Three floating wind
turbine substructure designs, a semi-submersible, a spar-buoy and the Hexafloat concept proposed by Saipem are compared in
multiple environmental conditions involving hundreds of simulations. The considered codes include TU Berlin's QBlade,
NREL's OpenFAST and Principia's DeepLines. Statistics, extreme and fatigue loads of key load sensors are discussed.
OpenFAST and QBlade results were refined over the span of several months, correcting small bugs that may arise in such a
complex set-up and ultimately aligning the models better. DeepLines has not benefitted from such improvements due to budget
and time limitations, which explains the poorer agreement noted for this code in many instances. These results are nevertheless
included as they are representative of what could be achieved with limited time and budget often connected to industrial
processes.
The statistical comparison revealed good agreement between the codes in their ability to predict general system dynamics.
Nonetheless some differences, particularly in the coupling with the controller, emerged. Blade pitch – floater pitch self-excitation
is noted in the Softwind and Hexafloat designs. While this phenomenon is present in all three codes, it is more accentuated in
QBlade, despite all three sharing the blade pitch controller logic. A possible explanation for this phenomenon was put forward
by the authors in the first part of this study (Behrens De Luna et al., 2023) and is linked to larger variations in rotor speed in

QBlade. Above rated wind speed, such variations cause the pitch controller to intervene more aggressively, thus triggering the floater pitch instability. Upon further investigation, aerodynamic torque is found to be more sensitive to blade pitch variations at low wind speeds in QBlade, which causes the response of the coupled turbine and controller system to be faster and thus more prone to instability. This self-excitation is found to be the cause of increased fore-aft tower base and out-of-plane root bending moment lifetime DELs in QBlade in both the Hexafloat and Softwind designs and demonstrated how small differences in modeling can have a significant impact on design loads.

No clear trend is noted when ultimate loads are compared. Taking QBlade as a reference point, ultimate loads are regularly found to be in the $\pm15\%$ range, with only some exceeding it. Although not discussed in detail in this work, part of these differences stem from the fact that the compared ultimate loads are selected according to the so-called *"mean of max"* method according to international standard indications (IEC61400-1, Annex G). As shown in (Papi et al., 2023), small differences in ultimate loads may cause the method to select a different maximum, amplifying the difference between the models. In addition, the different FOWT designs have a different dynamical response to the environmental conditions, thus affecting the ultimate loads differently. Fatigue loads, namely lifetime DELs, show a clear trend: OpenFAST generally predicts higher loads than QBlade, while DeepLines predicts lower lifetime fatigue loads. The reason for the latter being a different model set-up of the Softwind design in DeepLines and the lower effect of the blade pitch-platform pitch instability in the Hexafloat design. The exception to this is represented by tower base lifetime DELs, which for the Softwind design, are lower in OpenFAST. The root cause of this behavior in the Softwind design is again the floater pitch – blade pitch interaction, which is higher in QBlade compared to the two other codes. The higher DELs in OpenFAST are in line with other authors' findings, who observed higher fatigue loads in BEM-based codes compared to in LLFVW-based codes. In this study however, OpenFAST differs from the other two codes also in the structural modeling: the former utilizing a modal structural model without the ability to model blade torsion while the latter two feature a multi-body model that includes blade torsion. Despite the trend being consistent between the codes, the magnitude of the lifetime DEL overestimation is different in the two designs where OpenFAST and QBlade are compared, OC4 and Softwind. In fact, in Softwind, blade root DELs are 2% to 14% higher in OpenFAST, while in OC4 they are up to 1.5% higher. The analysis of CPSDs highlighted greater response at the 1P frequency in OpenFAST in the former design, while in OC4 the main difference between OpenFAST and QBlade is mostly confined to higher response in OpenFAST at very low frequencies. This low frequency difference is driven by increased rotor speed variation, in turn caused by differences in aerodynamic modeling.

In conclusion, the relatively simpler model assumptions adopted in OpenFAST are found to be able to reproduce the system dynamics adequately for the considered designs. No clear trend is noted for extreme loads. In fact, these differences could not be traced back to a specific engineering model or modelling choice. In this regard, including a larger set of extreme load cases with more parameter variations could help give a clearer picture of the differences in ultimate loading between the codes and the FOWT designs. On the other hand, a clear trend is noted in fatigue loads. This may be explained by the difference in aerodynamic models, in particular the comparison between the BEM-based OpenFAST and the LLFVW-based QBlade is consistent with existing scientific literature. DeepLines however contradicts this trend. While this may be, at least in part, due to setup differences

in the Softwind design and to this code being less prone to blade pitch-floater pitch self-excitation, this aspect is identified as a key point for future research.

Overall, the main outcomes of this study can be summarized as follows: i) the differences between the compared modelling theories are consistent with the existing body of literature on onshore wind turbines. ii) the greater movement that FOWTs are allowed did not exacerbate the differences to the point that simpler models, such as OpenFAST, are outdated. These tools remain reliable for extreme load estimation. For fatigue loads, underestimation with respect to more physically accurate theories of 2-15% depending on the specific load sensor can be expected. Therefore, within the limitations highlighted in this and other similar works, these models are still relevant for industry and for many research applications.

**Nomenclature**

COD      Co-Directional

CPSD    Cumulative Power Spectral Density

CS        Coordinate System

DLC     Design Load Case

$E[\varepsilon_1| \varepsilon_2]$ Expected value of $\varepsilon_1$ conditioned on $\varepsilon_2$

ECD     Extreme Change of Direction with coherent gust

ESS      Extreme Sea State

ETM     Extreme Turbulence Model

EWM    Extreme Wind Model

FOWT    Floating Offshore Wind Turbine

MUL     Multi-Directional

NSS      Normal Sea State

NTM     Normal Turbulence Model

OC4     OC4 DeepCWind semi-submersible

PSD      Power Spectral Density

Sims.     Simulations

SSS      Severe Sea State

$H_S$       Significant Wave Height (m)

$T_P$        Peak Spectral Period (s)

$M_{WW}$    Mean Wind-Wave misalignment (°)

$U_W$      Wind Speed

$V_{in}/V_{out}$ Cut-in/Cut-out wind speed (m/s)

ws       Wind speed

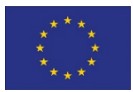 **Funding** This work has received support from the FLOATECH project, funded by the European Union's Horizon 2020 research and innovation programme under grant agreement No. 101007142

**Data Availability** The simulation results used in this study are publicly available at 10.5281/zenodo.7254241. The met-ocean conditions are also available at doi.org/10.1088/1742-6596/2385/1/012117. The QBlade-Ocean models upon which the models tested herein are based are available at 10.5281/zenodo.6397352 (OC5), 10.5281/zenodo.6397358 (Softwind), 10.5281/zenodo.6397313 (Hexafloat) and the modifications required to align them with the models tested herein are detailed in 10.5281/zenodo.7817707.


**Competing Interest** At least one of the (co-)authors is a member of the editorial board of Wind Energy Science. The peer-review process was guided by an independent editor, and the authors also have no other competing interests to declare.

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

886

## 6 Appendix A – Minimum Ultimate Loads

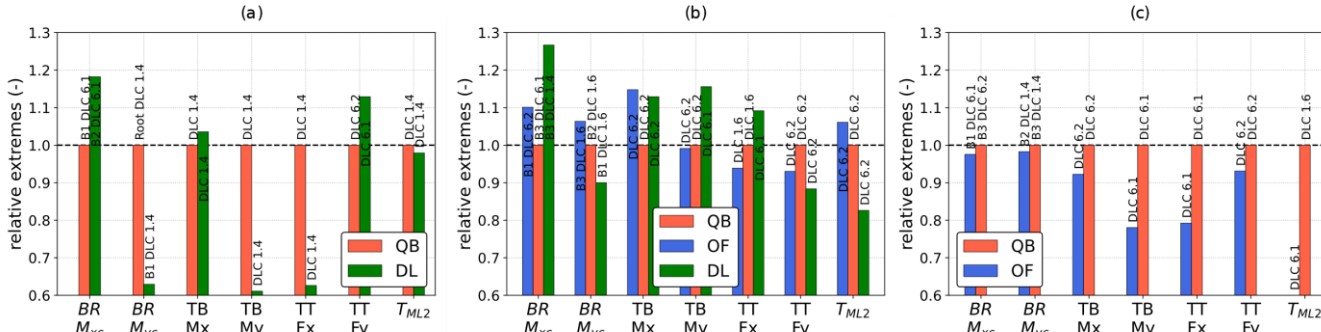

**Figure A1: Selection of ultimate loads (minimum) recorded in the three simulation codes. (a) DTU 10MW Hexafloat, (b) DTU 10MW Softwind and (c) NREL 5MW OC4.**