# Peer review of "Quantifying the Impact of Modeling Fidelity on Different"

_Wind Energy Science, 2023_

## Author Comment (AC1)

Dear Reviewers, dear Editor,

thank you for your time managing and reviewing our work and for your feedback. Based on the Reviewers' suggestions, we have done our best to improve the paper. Before answering your observations in detail we would like to point out that we decided to change the title of this study, from the original *"A Code-to-Code Comparison for Floating Offshore Wind Turbine Simulation in Realistic Environmental Conditions: Quantifying the Impact of Modeling Fidelity on Different Substructure Concepts"* to *"Quantifying the Impact of Modeling Fidelity on Different Substructure Concepts for Floating Offshore Wind Turbines - Part II: Code-to-Code Comparison in Realistic Environmental Conditions"*. We believe this change better links this paper with WES-2023-117, also under review in the WES Journal. WES-2023-117 is in fact "Part I" of this two-part study and lays the groundwork for the considerations done in this study. We hope this change also helps address some of Reviewer's #1 concerns regarding our citation of WES-2023-107 to explain some of the differences we observed in this paper. We believe we should have presented the two papers as a two-part study from the start, as they are deeply linked and part of the same project.

We have provided detailed answers to your comments below, in blue colored text for your convenience.

Best regards,

F. Papi, G. Troise, R. Behrens de Luna, J. Saverin, S. Perez-Becker,
D. Marten, M.L. Ducasse, A. Bianchini

ooo   ooo   ooo

**Reviewer #1 comments:**

The paper presents an interesting and comprehensive study on the load and performance of three floating offshore wind turbines, considering a very wide range of operating conditions, constructed by resorting to a well documented data base of ocean conditions (and already used in previous studies). Three different HAWTs are considered, each one featuring a different floating foundation system. Three different simulations packages, covering two main fidelity levels (BEM and LLFVW), are used to simulate the movement and the load acting on the turbines.

The paper highlights both the effects of the floating operation on the turbine behavior and load as well as the capabilities of the codes to predict such behaviors. In the end, the paper documents a study which is fully relevant for the WES journal and certainly of archival value.

Despite the general positive evaluation of the paper, in my opinion the document could be improved. In the following I provide some remarks that the authors may consider in their review.

Thank you for the constructive feedback and for the general appreciation of our work. Based on your suggestions, we have substantially re-worked the manuscript. Please find below our detailed answers to the specific comments you raised.

1. In general, the paper is difficult to read; it is very long and there is an exaggerated use of acronyms, which complicates the comprehension of the text in several instances. Some sections may also be divided in subsections or, at least, to be divided in more paragraphs. I invite the authors to find a way to make the discussion more synthetic and more effective or schematic, eliminating what is not strictly necessary for the paper (maybe referring to the reports of FLOATECH project). In the results section of the paper, for example, most of the comments are descriptions of the figures and of the differences between the codes, and less space is dedicated to the explanation of the observed differences.

Thank you again for the constructive feedback. This work is the summary of a long and – in our opinion – challenging research project. As such, the first version of this document reflected such complexity and, as the Reviewer pointed out, was probably excessively long and complex. To address this, the results section of the paper was completely re-worked. In particular, we divided the extreme results analysis into more sections (L365 to L413 of revised manuscript). We also streamlined discussion of fatigue loads (L459 to L544 of revised manuscript) by removing Fig. 9 and 12, as they were not essential to the discussion of the differences between the codes. We simplified Figs. 19 and 20, merging them into one and re-worked the tower base fatigue loads discussion. Finally, we reduced the use of acronyms to improve readability. Overall, the main body of text more focused on the code-to-code differences. We hope this change meets the Reviewer's expectations.

2. The conclusions are, in some aspects, weak. For example, the self-excitation phenomenon (one of the most interesting findings of the paper) is predicted in different way by the codes, but the explanation of the over-estimate of Q-Blade code is demanded to another paper (Behrens de Luna et al.), still unpublished; it could have been a relevant discussion and conclusion also of the present paper. As a second example, the results often show differences between the predictions of OpenFAST and DeepLines Wind, despite the two codes are of the same fidelity level; the explanation of these results are demanded to future studies, while one would have expected such discussion in this paper, for example by resorting to the different structural model of OpenFAST or to the compiling/import issues of DeepLines Wind mentioned in the paper (and see below for the remarks on these aspects).

The Reviewer stressed an important point. The conclusion of this study is indeed different from our initial expectation when we started this work and the FLOATECH proposal was written. Before going into this, we were expecting that the additional motion afforded by the floating installation would accentuate the differences found by many between aerodynamic and structural models for onshore wind turbines. Throughout this study, however, we have mostly found small differences between the compared codes. We have tried to highlight this better in the conclusions (L584-588). Regarding the explanation of the self-excitation phenomenon, the paper by Behrens de Luna et al. that we referenced is part of the same work that led to this paper. Based on the reviews we received on both papers, we have decided to change the title of both works to link them together and hopefully improve clarity and scope of both works. Based on the intuition in that paper, we have run additional analyses in OpenFAST and QBlade to shed some light on the possible causes of the phenomenon. Various physical phenomena could, in principle, cause such a difference in excitation. However, by process of exclusion, differences in hydrodynamic excitation are unlikely to be the cause of the increased self-excitation in QBlade, as nearly identical response in QBlade and OpenFAST was noted at the Softwind's pitch natural frequency in part one of this study ([3], Fig. 13). Moreover, the aerodynamic model is also not the cause, as switching to DBEM in QBlade did not improve agreement in this regard with respect to OpenFAST (not shown herein for brevity). As stated previously, OpenFAST doesn't include blade torsion. However, switching to a rigid structure did not improve the agreement of OpenFAST and QBlade. A possible explanation for the difference in blade pitch – platform pitch self-excitation was put forward in part one of this study [3] and is related to increased aerodynamic torque variation in QBlade with respect to the other two codes. Indeed, upon further investigations, differences in the system dynamics, and how they interact with the control system, could explain the observed behavior. As explained in detail by Abbas et al. [2], the controller and turbine can be seen as a closed-loop second-order system, characterized by a natural frequency at a certain operating wind speed:

$$\omega^2 = k_i(U_{op})B = k_i(U_{op})\frac{N_g}{J}\frac{\partial \tau_a}{\partial \beta} \qquad (1)$$

where $N_g$ and $J$ are the gearbox ratio and rotor inertia, which are the same in OpenFAST, QBlade, and DeepLines. The higher the natural frequency, the more responsive the system is to an external disturbance such as a platform pitch oscillation. The integral controller gain $k_i$ is also the same in the two codes, as it depends on the controller tuning. The slope of the aerodynamic torque as a function of blade pitch is, however, different in the two codes. The derivative of aerodynamic torque as a function of blade pitch for the mean 11 m/s operating conditions is shown in Fig. 12 (b). As $\frac{\partial \tau_a}{\partial \beta}$ is larger in magnitude for QBlade at the mean operating blade pitch of approximately 0.5°, from eq. 1, $\omega^2$ is also larger, leading to increased self-excitation in QBlade.

[Figure]

Figure 12: (a) aerodynamic torque as a function of blade pitch for OpenFAST and QBlade for 11 m/s operating TSR, and relative trendlines. (b) derivative of aerodynamic torque as a function of blade pitch computed from analytic derivative of trendlines.

This explanation is now included in the paper (L495-518).

Finally, we are aware of the differences between DeepLines and OpenFAST, despite them being of a similar fidelity level. We have refined the OpenFAST and QBlade results over the span of several months, correcting small bugs that may arise in such a complex set-up and ultimately aligning the models better. DeepLines has not benefitted from such improvements. To this end, we have debated internally whether or not to include DeepLines results. We have ultimately decided to include them despite the set-up issues explained in Section 2.3.2, because general overall agreement with the open-source codes is good and this result is representative of what an industrial partner could achieve with the limited time and budget often connected to industrial processes. We have once more highlighted this better in the conclusions. (L498-522 of revised manuscript)

Detailed comments:

- In section 3.2, the reason for using a simplified structural model in OpenFAST is not convincing; if one wants to see the effect of using a simplified structural model, a comparative analysis must be done changing this particular model within the same simulation framework, and not introducing a further variability in a context of code-to-code comparison. The implications of this choice should be better highlighted in the results section.

In our view, this study is a high-level code-to-code comparison aimed at assessing the impact of multi-fidelity modelling choices in the three codes with respect to each other. In the case of structural modelling in OpenFAST, we could have chosen indeed to use BeamDyn, which is able to model blade torsion in addition

to flapwise and edgewise deformation of the blade and is more theoretically sound especially in the case of pre-bent blades. However, the three structural models in the three codes would still have differed, in both theory and implementation. Therefore, we chose to use the simpler approach in OpenFAST, with the intention of investigating the global implications of the overall modelling choices in each code. We agree that the wording used in section 3.2 may be misleading and thus rephrased in the revised manuscript (L254-255 of revised manuscript).

 - At the beginning of Section 4, it is mentioned that not all the simulation runs reached convergence in OpenFAST or DeepLines Wind. While this might be expected, it would be interesting to briefly discuss what are the reasons for these failed convergences.

Thank you for the comment. Indeed, disclosing more information in this regard may help others facing similar issues in their simulations. One simulation in OpenFAST did not converge. Upon detailed inspection of the result, the issue seems to be related to numerical instabilities in the structural solver. In DeepLines-wind the issue can be traced back to instabilities in the numerical integration scheme. Despite an initial attempt to solve these issues in both codes, we ultimately did not have the resources to attempt to fine-tune the numerical parameters in the two codes and solve the issues. The beginning of Section 4 has been edited to reflect these changes (L310-314 of revised manuscript).

 - In section 4.1, line 307, a compilation issue is mentioned in DeepLines Wind, which has a relevant impact on the results. It is not clear, for the reader, what could be the practical consequences of this issue: if the compilation issue could be easily solved by the Authors, the technical relevance of the results obtained with DeepLines Wind is questionable; if, instead, the issue is an inherent feature of the code, a general improvement to the code is needed. Please explain.

DeepLines Wind uses a different convention for pitch angle than OpenFAST and QBlade. Thus, the ROSCO controller needed to account for this to be coupled to the former. In the re-compiled version, minimum rotor speed is not enforced. This can be very clearly seen in Fig. 3 (b, e). To our best knowledge, the control routine is the same as that used in OpenFAST and QBlade for wind speeds higher than 7 m/s. The discrepancy has a relevant impact on fatigue loads, while we believe it does not affect extreme loads as only low wind speeds are affected. This is acknowledged throughout the results section of the paper. We realize that this is not ideal, but there is no way for us to rectify this issue without removing DeepLines Wind entirely from the comparison, which we would prefer not to do as we believe it is a worthy addition to the code-to-code comparison. We changed the text in section 4.1 to better highlight the influence and causes of this discrepancy (L334-341 of revised manuscript).

- In section 4.2, line 374, an issue of DeepLines Wind in importing the wind field is mentioned. Again, is it an inherent limitation of the code with respect the other ones, or was it a simple issue in the set-up that could have been solved by the Authors?

In medium-fidelity wind turbine simulation tools such as those used in this study wind fields are imported as three-dimensional wind boxes, where the first two dimensions are the height and width of the field and the third is time. Wind boxes are often shifted on import of a time equals to R/U so that the turbine is fully immersed in the wind box even in case of yaw misalignment. Despite using the same wind fields, differences on import ultimately cause the wind fields to be shifted in DeepLines-Wind. This is not a limitation of the examined codes, nor something that we could address with set-up changes. We have added a paragraph to section 4.2 (L405-407 of revised manuscript) to better reflect this.

- Figure 10 (and following similar ones): personally I do not fully appreciate the use of 'cumulative PSD' in the plots, but I recognize its effectiveness with respect to the standard PSD diagram; for clarify, I recommend to specify in the paper, when commenting Figure 1 for the first time, how to read a cumulative PSD diagram.

Thank you for the feedback. Cumulative PSDs are the most effective way we found to showcase the differences between the codes. We agree that they can be confusing for readers that are not accustomed to such metric. We added an explanation on how to read CPSDs when introducing them in Fig. 9 (L459-462 of revised manuscript).

- Figure 20: a significant difference exists between the Q-blade and DeepLines Wind in the frequency range 0.4-0.6 Hz, the authors are encouraged to comment this aspect in the paper.

Indeed, there is a difference in this frequency range in the tower base fore-aft bending moment of QBlade and DeepLines wind. In this frequency range, a fore-aft mode of the Hexafloat structure, where the upper floater structure oscillates 180° out-of-phase with the rotor collective flapwise mode. Although we were unable to find the root cause of this difference, the difference between the two codes is most likely liked to them capturing this mode differently. It must be noted that the difference is amplified by the semi-log scale of Fig. 20 and is in reality very small.

°°°°°°°°°°°°°°°°°°°°°°°°°°°°°°°°°°°°°°°°°°°°°°°°°°

[1] Papi, F., Bianchini, A., Troise, G., Mirra, G., Marten, D., Saverin, J., Behrens de Luna, R., Ducasse, M.-L., and Honnet, J.: Deliverable 2.4  Full report on the estimated reduction of uncertainty in comparison to the state-of-the-art codes OpenFAST and DeepLines Wind, 2023.

[2] Abbas, N. J., Zalkind, D. S., Pao, L., and Wright, A.: A reference open-source controller for fixed and floating offshore wind turbines, Wind Energy Science, 7, 53–73, https://doi.org/10.5194/wes-7-53-2022, 2022.

[3] Behrens De Luna, R., Perez-Becker, S., Saverin, J., Marten, D., Papi, F., Ducasse, M.-L., Bonnefoy, F., Bianchini, A., Nayeri, C. N., and Paschereit, C. O.: Verifying QBlade-Ocean: A Hydrodynamic Extension to the Wind Turbine Simulation Tool QBlade, Wind Energy Science Discussions, 1–36, https://doi.org/10.5194/wes-2023-117, 2023.

---

## Author Comment (AC2)

Dear Reviewers, dear Editor,

thank you for your time managing and reviewing our work and for your feedback. Based on the Reviewers' suggestions, we have done our best to improve the paper. Before answering your observations in detail we would like to point out that we decided to change the title of this study, from the original *"A Code-to-Code Comparison for Floating Offshore Wind Turbine Simulation in Realistic Environmental Conditions: Quantifying the Impact of Modeling Fidelity on Different Substructure Concepts"* to *"Quantifying the Impact of Modeling Fidelity on Different Substructure Concepts for Floating Offshore Wind Turbines - Part II: Code-to-Code Comparison in Realistic Environmental Conditions"*. We believe this change better links this paper with WES-2023-117, also under review in the WES Journal. WES-2023-117 is in fact "Part I" of this two-part study and lays the groundwork for the considerations done in this study. We hope this change also helps address some of Reviewer's #1 concerns regarding our citation of WES-2023-107 to explain some of the differences we observed in this paper. We believe we should have presented the two papers as a two-part study from the start, as they are deeply linked and part of the same project.

We have provided detailed answers to your comments below, in blue colored text for your convenience.

Best regards,

*F. Papi, G. Troise, R. Behrens de Luna, J. Saverin, S. Perez-Becker,*
*D. Marten, M.L. Ducasse, A. Bianchini*

∘∘∘   ∘∘∘   ∘∘∘

**Reviewer #2 comments:**

The manuscript discusses the performance of floating offshore wind turbines. For that, the differences in modelling these facilities are studied using three different simulation codes and three substructure designs and considering realistic environmental conditions. Furthermore, the study is made within the EU project floatech. The authors find that while all codes provide similar results for the system dynamics, differences are observed in the estimation of fatigue loads. I find the paper well written and the analysis sound for the offshore wind turbine community. I therefore consider it is suitable for publication in Wind Energy Science journal, provided the authors address the following minor comments:

Thank you for your time and valuable feedback. We are glad that the Reviewer generally appreciated our work. We provide detailed answers to the points raised below.

- The environmental conditions used for the present work correspond to the Scottish island of Barra. The authors argument this choice because, among other arguments, this location presents harsh conditions. The question arises therefore about how representative is this study of more standard offshore environmental conditions. I would appreciate if the authors can address this point in the manuscript.

The Reviewer raises an interesting point. It is hard to give a general and definitive answer to this without any additional research. As stated in the manuscript, we have selected these environmental conditions partly because they are believed to be particularly severe, and thus to better highlight the potential differences

between the codes. In other words, we were looking for environmental conditions that would give us strong wind and wave excitement on the turbines. Wind and wave actions are coupled in a FOWT, in the sense that one excitation influences the other and vice-versa. We were looking to maximize this interaction to comprehensively test QBlade-Ocean and highlight any differences in the multi-fidelity models. In more "standard" environmental conditions we would expect less wind-wave coupling in the FOWT dynamics and thus less differences between the codes. This reasoning is now reflected in the paper in section 2.1 (L104-106 of revised manuscript). Again, it is hard to state this with absolute certainty without performing additional research on the topic.

- Table 1 caption should describe the variables in more detail. Overall, while in another work from the authors this dataset is discussed, more detail about it would help the reader of the present manuscript.

Table 1 is indeed very dense and can be hard to understand for researchers out of the field of load simulation. We then improved the caption of the table as suggested, but also added additional details and references to section 2.2, where the interested reader can find more information (L118-120 of revised manuscript).

- Why DLC 1.4 simulations are only 10 minutes long? What is different with respect to the other ones?

The Reviewer's comment is on point. We did not explain this in detail in the first draft to keep the discussion short, but the choice indeed needs justification. In DLC 1.4 ad extreme wind gust with direction change occurs around 100 second into the simulation, which causes the turbine to shut down. Because this is an extreme load DLC, we are interested in the extreme loads that arise as a consequence of this event. These extreme loads occur in correspondence of the event itself or right after. If the Reviewer is interested, the timeseries in Fig. 42 in [1] show this nicely. As such, we don't need to simulate a full hour of operation in this DLC. We added an explanation to clarify this point in section 2.2 (L122-128 of revised manuscript).

- In figure 2, the turbine models from left to right do not agree with the order in their description in the sections below. This is a very minor issue but the authors may want to correct the inconsistency.

Thank you for pointing this out. We agree that this suggestion helps to keep the paper consistent and schematic. We changed the order of the models in Fig. 2 and the description of the figure to boot.

- I agree with the other Reviewer that the lack of convergence in some simulations in OpenFAST and DeepLine deserves a deeper discussion.

This should have been explained better in the first draft as requested by both Reviewers. Indeed, disclosing more information in this regard may help others facing similar issues in their simulations. One simulation in OpenFAST did not converge. Upon detailed inspection of this specific result, the issue seems to be related to numerical instabilities in the structural solver. In DeepLines-Wind the issue can be traced back to instabilities in the numerical integration scheme. Despite an initial attempt to solve these issues in both codes, we ultimately did not have the resources to attempt to fine-tune the numerical parameters in the two codes and solve the issues. The beginning of section 4 has been edited to reflect these changes (L310-314 of revised manuscript).

- While the manuscript reads well, it presents several typos that should be corrected.

Thank you again for pointing this out. We have proofread the manuscript once over and hopefully have rectified all the issues.

[1] Papi, F., Bianchini, A., Troise, G., Mirra, G., Marten, D., Saverin, J., Behrens de Luna, R., Ducasse, M.-L., and Honnet, J.: Deliverable 2.4  Full report on the estimated reduction of uncertainty in comparison to the state-of-the-art codes OpenFAST and DeepLines Wind, 2023.

[2] Abbas, N. J., Zalkind, D. S., Pao, L., and Wright, A.: A reference open-source controller for fixed and floating offshore wind turbines, Wind Energy Science, 7, 53–73, https://doi.org/10.5194/wes-7-53-2022, 2022.

[3] Behrens De Luna, R., Perez-Becker, S., Saverin, J., Marten, D., Papi, F., Ducasse, M.-L., Bonnefoy, F., Bianchini, A., Nayeri, C. N., and Paschereit, C. O.: Verifying QBlade-Ocean: A Hydrodynamic Extension to the Wind Turbine Simulation Tool QBlade, Wind Energy Science Discussions, 1–36, https://doi.org/10.5194/wes-2023-117, 2023.

---

## Author Response (AR2)

**WES-2023-107 Response to Editor**

Dear Editor,

thank you for the time you spent managing and reviewing our work, providing highly qualified and detailed comments that helped us fix some flaws of our study. Please find below detailed answers to your comments, reported in blue colored text for your convenience.
Best regards,

*F. Papi, G. Troise, R. Behrens de Luna, J. Saverin, S. Perez-Becker,*
*D. Marten, M.L. Ducasse, A. Bianchini*

ooo   ooo   ooo

The article is mostly well written.

It seems more suitable as a technical report, but could in principle be published. However, I would ask the authors to consider what readers can actually learn from this work, and to add a few sentences about this to the manuscript.

Dear Editor, we have addressed your relevant comment in two ways:

1. Summarized the objective of this work better in the introduction of the paper: "Ultimately, the objective of this work is to provide wind turbine modelers and practitioners with a quantitative indication of the impact that model fidelity has on FOWT design loads and provide guidance in the selection of the most suitable approach for each task at hand."
2. Improved the description of the main handouts of this work in the conclusions: "Overall, the main outcomes of this study can be summarized as follows: i) the differences between the compared modelling theories are consistent with the existing body of literature on onshore wind turbines. ii) the greater movement that FOWTs are allowed did not exacerbate the differences to the point that simpler models, such as OpenFAST, are outdated. These tools remain reliable for extreme load estimation. For fatigue loads, underestimation with respect to more physically accurate theories of 2-15% depending on the specific load sensor can be expected. Therefore, within the limitations highlighted in this and other similar works, these models are still relevant for industry and for many research applications."

That said, there are also a number of minor issues that should still be addressed:

1. Abstract - "Consensus is arising on considering floating offshore wind as the most promising technology to increase renewable energy generation offshore." - Said like that, this does not seem to be true. It is still monopile foundations where we expect the biggest increases in the coming years, in

absolute terms (e.g. see US DoE Offshore Wind Market Report). Either modify or remove the sentence, or provide justification for it.

We agree with the Editor in the fact that this statement, in its current form, could be misleading. We have reformulated as follows:
"Floating offshore wind is widely considered as a promising technology to harvest renewable energy in deep ocean waters and increase clean energy generation offshore."

2. Abstract - "differences in fatigue loads are larger for tower base loads than for blade root loads, due to the larger influence substructure dynamics have on these loads." - I do not see justification for this claim in the paper, where is this shown exactly? Why is it not simply the larger moment arm, for example, that explains larger differences in loads?

Thank you for pointing this out. Indeed, we did not highlight this aspect properly in the previous version of the paper. Separating the causes of the differences can be challenging in a scenario like this. Notwithstanding this, there are some indications throughout the paper that indeed substructure movement influences the differences between the codes. We have now better highlighted this in two separate parts of the study. First in section 4.1, where standard deviation of platform pitch is shown in Fig. 4: "Platform pitch is remarkably similar in mean value, standard deviation, and minimum/maximum value for the OC4 test-case (Fig. 4 (e)), although higher standard deviation can be noted for wind speed near cut-in and cut-out in OpenFAST. This is interesting because a higher platform-pitch standard deviation indicates increased gravitational and inertial loading variations on the tower. Very good agreement between OpenFAST and QBlade is also shown in Fig. 4 (c). Despite platform pitch standard deviation being lower in QBlade for most wind speeds, at 13 m/s mean wind speed it is higher for QBlade.".

Then also in section 4.3.1, where tower and mooring lifetime DELs are shown: "However, the differences in fore-aft tower lifetime DELs (TT Fx, TB My) in Fig. 14 (a) and (c) are larger than those in out-of-plane blade root lifetime DELs in Fig. 8 (a) and (c). Such phenomenon can be traced back, at least in part, to the differences in platform pitch that are noted in Fig. 4, which cause larger or smaller variations in gravitational and inertial forces on the tower, increasing the difference in tower lifetime DELs. "

3. line 36: "on-going OC6" - As far as I know, OC6 has been concluded and OC7 is now on-going.

This is correct, thank you for pointing it out. The statement was true at the date of submission of the draft. It has been corrected as follows: "[…] OC6 (Jonkman and Musial, 2010; Robertson et al., 2014b, 2017; Bergua and et. al., 2023) and the on-going OC7."

4. line 95: "we did not find suitable met-ocean conditions for this analysis" - This is astonishing, given the large effort on this and many other projects dealing with similar issues. For example, the FP7 MARINE project developed such data (DOI: 10.1115/1.4029842). Why are these data not sufficient for this code-to-code comparison?

The Editor comment was again right, as the text was too concise to be accurate. The analysis required a large amount of specific environmental data, and, as the Editor is surely aware, international

standards are fairly specific regarding the requirements. In addition to the long-term statistical descriptions of the installation sites we also need environmental contours of wind speed and significant wave height. We were aware of the work that the Editor mentions, but, as far as we are aware it does not contain environmental contours in the form that we needed in this study. Moreover, wind-wave misalignment is not considered in the statistical representations in (DOI: 10.1115/1.4029842). Therefore, we decided to follow the route highlighted in the paper. We have amended the paper as follows:

"Some databases containing such met-ocean data can be found in previous work – for a comprehensive literature review see (Papi and Bianchini, 2023) – however if we restrict our research to Europe, we did not find met-ocean conditions that were completely suitable for this analysis. In fact, although conditions for some reference European sites can be found in the open literature, such as in (Li et al., 2015), specific environmental contours are required to perform ultimate load calculations according to the prescriptions of International Standards. Therefore, an open-source procedure to obtain and prepare long-term environmental data so it can be used in a design load calculation of an offshore wind turbine was developed. The procedure that is detailed in (Papi et al., 2022c) and is available open-access for others to use and improve upon (10.5281/zenodo.6972014). A highlight of the procedure is the fact that the statistical description of the installation site includes wind-wave misalignment, which has been shown to have a significant effect on loading (Stewart, 2016)."

5. line 138: Are half-hour simulations long enough to allow for low-frequency responses to build up to realistic values? The usual simulation times used, mainly because of this issue, are three hours. Please explain why 30 minutes is sufficient here!

Thank you for pointing out this important point. We have debated this internally a lot during the project. Some of the numerical models we ran, such as the LLFVW in QBlade, although manageable with modern hardware, are computationally more expensive than most engineering tools that are currently being used for this sort of application, such as OpenFAST. To keep computations reasonable within this project, we tried to reduce the simulation burden as much as possible. The choice of running half-hour simulations instead of the more common one-hour or three-hour intervals was made for this reason. However, this choice is backed-up by existing research. Based on the Editor's observation, we have included the following explanation in the paper: "The half-hour simulation length differs from the more commonly used one or three-hour simulation lengths. The rationale for such long simulations is to allow for enough time for low-frequency response, typical of FOWT systems, to build-up. Existing research (Stewart, 2016) however, indicates that the total time that is simulated within each environmental bin is the most important factor for fatigue-load estimation, rather than the length of each simulation. Moreover, based on the results in (Stewart, 2016), increasing simulation time beyond half-hour for each environmental bin does not appear to yield improved fatigue estimations in most cases. Therefore, considering the comparative nature of the study, two half-hour simulation for each environmental bin were considered sufficient for fatigue load comparison."

6. Table 1: Explain abbreviations (ws, sims) in title.

Thank you for pointing this out. We have added these abbreviations to the nomenclature list and amended the description of table 1.

7. line 294: "QBlade is able to model Wheeler wave stretching" - This should also be available in all codes that implement linear wave theory! And was it used, or not?

To the best of our knowledge, Wheeler wave stretching is currently not available in OpenFAST. On the other hand, after double checking with the project member that ran the DeepLines simulations, we can confirm that Wheeler wave stretching was included in those simulations. The paper was corrected to reflect this (L308).

8. Fig. 7: Why are the blades pitched differently? And why is there no turbulence in the DeepLines model, but in the other two simulations?

Thank you for pointing this out. Simulations in Fig. 7 feature wind speed above cut-out. The controller is not active in these simulations, and the rotor is left to idle with the blades feathered. When the controller is not active, we could not export rotor speed, blade pitch and wind speed from DeepLines. Placeholder values were added to the dataset instead: constant values of 0 for blade pitch and rotor speed and constant value equal to the mean wind speed value for wind speed. We have removed the timeseries in question, as they add confusion to the figure, and cleared this in the figure description.

9. References, line 852: This is incomplete. Year is missing, type of publication and lead institution is missing. And what is "n.d." supposed to mean?

Thank you again for the thorough review. The reference was indeed incomplete. We have added the missing items.

Please also correct a number of tpyos:

1. line 119: "selected process" -> "selection process"
2. line 298: "hydrodunamic" -> "hydrodynamic"
3. line 332: "DeepLines respect to" -> "DeepLines with respect to"
4. line 355: "shown in Fig. 4 (b)" -> "shown in Fig. 4 (d)" ?
5. line 416: "platform roll" -> "platform pitch" ?
6. line 523: "are shown for the in Figure 12" -> "are shown in Figure 12"
7. Figs. 15 and 18: "flier values" -> "outlier values"
8. line 630: "Fig. 18 (b,c)" -> "Fig. 18 (a, b)" ? (Fig. 18 does not have a panel c)
9. line 678: "for extreme loads.. In fact, " -> "for extreme loads. In fact, "

Thank you for pointing all these points out. They all have been corrected. In the case of typo #5, an additional clarification was also added to the text. In case of typo #8, the label "c" was added to the figure where it was missing.